

# Fundamental Oxidation Processes in the Remote Marine Atmosphere Investigated Using the NO-NO2-O3 Photostationary State

Simone T. Andersen[1*], Beth S. Nelson[1], Katie A. Read[1,2], Shalini Punjabi[1,2], Luis Neves[3], Matthew J. Rowlinson[1], James Hopkins[1,2], Tomás Sherwen[1,2], Lisa K. Whalley[2,4], James D. Lee[1,2], and Lucy J. Carpenter[1]

[1]Wolfson Atmospheric Chemistry Laboratories (WACL), Department of Chemistry, University of York, Heslington, York, YO10 5DD, UK.

[2]National Centre for Atmospheric Science (NCAS), University of York, Heslington, York, YO10 5DD, UK.

[3]Instituto Nacional de Meteorologia e Geofísica, São Vicente (INMG), Mindelo, Cabo Verde.

[4]School of Chemistry, University of Leeds, Leeds, LS2 9JT

[*]Corresponding author: simone.andersen@york.ac.uk



# 1 Abstract

The photostationary state (PSS) equilibrium between NO and $NO_2$ is reached within minutes in the atmosphere and can be described by the PSS parameter, $\varphi$. Deviations from expected values of $\varphi$ have previously been used to infer missing oxidants in diverse locations, from highly polluted regions to the extremely clean conditions observed in the remote marine boundary layer (MBL), and have been interpreted as missing understanding of fundamental photochemistry. Here, contrary to these previous observations, we observe good agreement between PSS-derived $NO_2$ ($[NO_2]_{PSS\ ext.}$) calculated from photochemical model predictions of peroxy radicals ($RO_2$ and $HO_2$) and measured NO, $O_3$, and $jNO_2$, and observed $NO_2$ in extremely clean air containing low levels of CO (< 90 ppbV) and VOCs. However, in clean air containing small amounts of aged pollution (CO > 100 ppbV), we observed higher levels of $NO_2$ than inferred from the PSS, with $[NO_2]_{Obs.}/[NO_2]_{PSS\ ext.}$ of 1.12-1.68 (25th-75th percentile) implying 18.5-104 pptV (25th-75th percentile) of missing $RO_2$ radicals. Potential $NO_2$ measurement artefacts have to be carefully considered when comparing PSS-derived $NO_2$ to observed $NO_2$, but we show that the $NO_2$ artefact required to explain the deviation would have to be ~ 4 times greater than the maximum calculated from known interferences. If the missing $RO_2$ radicals have an ozone production efficiency equivalent to that of methyl peroxy radicals ($CH_3O_2$), then the calculated net ozone production including these additional oxidants is similar to that observed, within estimated uncertainties, once halogen oxide chemistry is accounted for. This implies that peroxy radicals cannot be excluded as the missing oxidant in clean marine air containing aged pollution, and that measured and modelled $RO_2$ are both significantly underestimated under these conditions.

# 2 Introduction

Tropospheric NO, $NO_2$ and $O_3$ are rapidly interconverted during the day via reactions (1-3), where NO is oxidised by $O_3$ into $NO_2$, which is then photolyzed into NO and $O(^3P)$, followed by a fast reaction of $O(^3P)$ with $O_2$ to return $O_3$.

$$NO + O_3 \rightarrow NO_2 + O_2 \tag{1}$$





$$NO_2 + hv \rightarrow NO + O(^3P) \qquad (hv \leq 410 \text{ nm}) \qquad (2)$$
$$O(^3P) + O_2 + M \rightarrow O_3 + M \qquad (3)$$

The photostationary state (PSS) equilibrium between NO and $NO_2$ is reached within

minutes (Leighton, 1961) if it is not impacted by fresh $NO_x$ emissions and if the photolysis rate
does not change quickly such as under rapidly changing cloud coverage (Mannschreck et al.,
2004). The photostationary state can be described by the Leighton ratio (Leighton, 1961) (eq.
I), where $jNO_2$ is the photolysis rate of $NO_2$ and $\varphi$ is the PSS parameter.
$$\varphi = \frac{jNO_2[NO_2]}{k_1[NO][O_3]} \qquad (I)$$

Under very polluted conditions, where $O_3$ is the only oxidant converting NO to $NO_2$, $\varphi$

is equal to 1 and the $NO_2$ at PSS can be estimated from the measured NO, $O_3$, and $jNO_2$ (eq.
II).
$$[NO_2]_{PSS} = \frac{k_1[NO][O_3]}{jNO_2} \qquad (II)$$

Deviations from $\varphi = 1$ suggest the presence of additional chemistry occurring (Calvert

and Stockwell, 1983), particularly the conversion of NO to $NO_2$ by reaction with an oxidant
other than $O_3$, such as hydroperoxy radicals ($HO_2$) and peroxy radicals ($RO_2$) (reactions 4-5,
where R represents any organic functional group) or with halogen oxides (IO, BrO; reactions
6-7) in the marine atmosphere.
$$RO_2 + NO \rightarrow RO + NO_2 \qquad (4)$$
$$HO_2 + NO \rightarrow OH + NO_2 \qquad (5)$$
$$IO + NO \rightarrow I + NO_2 \qquad (6)$$
$$BrO + NO \rightarrow Br + NO_2 \qquad (7)$$

By including these additional NO oxidation reactions, the $NO_2$ concentration at PSS

can be estimated using equation (III). The photostationary state of $NO/NO_2$ can also be used to
estimate the sum of $HO_2$ and $RO_2$ ($RO_x$) or the sum of BrO and IO (XO) in the atmosphere
using equation (IV) and (V) and assuming that $k_4 = k_5$ and $k_6 = k_7$, respectively:
$$[NO_2]_{PSS\ ext.} = \frac{(k_1[O_3] + k_4[RO_2] + k_5[HO_2] + k_6[IO] + k_7[BrO])[NO]}{jNO_2} \qquad (III)$$



$$[RO_2] + [HO_2] = \frac{j\text{NO}_2[\text{NO}_2] - (k_1[\text{O}_3] + k_6[\text{IO}] + k_7[\text{BrO}])[\text{NO}]}{k_{4,5}[\text{NO}]}$$ (IV)
$$[BrO] + [IO] = \frac{j\text{NO}_2[\text{NO}_2] - (k_1[\text{O}_3] + k_4[\text{RO}_2] + k_5[\text{HO}_2])[\text{NO}]}{k_{6,7}[\text{NO}]}$$ (V)

Previous studies reporting deviations in the PSS parameter to estimate $RO_x$

concentrations in the atmosphere are summarised in Table 1, which compares $[RO_x]_{PSS}$ against
measured and/or modelled $[RO_x]$. Measurements of $RO_x$ are predominantly conducted using
chemical amplification, where each $RO_2$ and $HO_2$ molecule in ambient air leads to the
formation of several $NO_2$ molecules by chain reactions caused by the addition of high
concentrations of NO and CO (Cantrell et al., 1993b). The resultant $NO_2$ can be detected and
converted back to a $RO_x$ concentration by quantification of the chain length of the reactions
via calibration, typically using known concentrations of $CH_3O_2$ or peroxyacetyl ($CH_3C(O)O_2$)
radicals (Cantrell et al., 1993b; Miyazaki et al., 2010; Wood and Charest, 2014). Since the basis
of the chemical amplification technique is detection of $RO_x$ radicals from their ability to oxidise
NO to $NO_2$ (reactions 4 and 5), which is also used to estimate $RO_x$ from the PSS, the $RO_x$
concentrations determined from these methods would be expected to agree reasonably well.
However, PSS-derived $RO_x$ concentrations are generally higher than both measured and
modelled values in rural conditions (Cantrell et al., 1997; Cantrell et al., 1993a; Ma et al., 2017;
Mannschreck et al., 2004; Volz-Thomas et al., 2003) with exceptions such as in the Pearl River
Delta where PSS-derived and measured $RO_x$ were comparable (Ma et al., 2017). During
campaigns in relatively clean regions with moderate influence from pollution (Amazon Basin
and Arabian Peninsula), PSS-derived $RO_x$ levels have been shown to be in good agreement
with modelled $RO_x$ (Tadic et al., 2020; Trebs et al., 2012). In the remote marine boundary layer
(MBL), PSS-derived $RO_x$ has been observed to be 1.27 times higher than the measured $RO_x$
over the South Atlantic Ocean, however, the measured $RO_x$ was approximately 4 times higher
than modelled (Hosaynali Beygi et al., 2011).

The difference between measured, modelled, and PSS-derived $RO_x$ can be due to a

variety of reasons. $RO_x$ concentrations calculated by box models rely on comprehensive
constraint from co-measured trace gases and a reaction scheme which accurately represents the
most important photochemical processes. Incomplete characterization of ambient trace gases
and/or reaction schemes can therefore result in uncertain $RO_x$ predictions. Large deviations
(factor of ~ 3) between modelled and measured $RO_x$ levels in a pine forest in the Rocky
Mountains were attributed to a combination of a missing photolytic source of $HO_2$ at midday


and a missing reaction forming $RO_2$ independently of sunlight in the model scheme (Wolfe et
al., 2014). PSS-derived $RO_x$ can be significantly over- or underestimated if the PSS has not
been established, for example due to rapidly changing photolysis rates or local sources of $NO_x$
(Mannschreck et al., 2004). Another reason for overestimation of PSS-derived $RO_x$ is $NO_2$
measurement artefacts (Bradshaw et al., 1999; Crawford et al., 1996), which results in
overestimated $NO_2$ concentrations. These are common in chemiluminescence instruments and
can be due to photolytic or thermal decomposition of HONO, peroxyacetyl nitrate (PAN), and
other nitrate molecules in the atmosphere (Bradshaw et al., 1999; Gao et al., 1994; Parrish et
al., 1990; Pollack et al., 2010; Reed et al., 2016; Ridley et al., 1988; Ryerson et al., 2000).

Measurements of $RO_x$ are also not without challenges due to effects from e.g. the high

reactivity of $RO_x$, humidity, non-linearity of the $NO_2$ detection, and formation of organic
nitrates and nitrites. In the first chemical amplification instruments, $NO_2$ was detected by
luminol chemiluminescence, which has a non-linear response to $NO_2$ resulting in the need for
a multipoint calibration (Cantrell et al., 1997).  However, more recent instruments use cavity
absorption phase shift (CAPS) (Duncianu et al., 2020; Wood and Charest, 2014), laser induced
fluorescence (LIF) (Sadanaga et al., 2004), or cavity ring-down spectroscopy (CRDS) (Liu and
Zhang, 2014) for detection of $NO_2$, all of which have been shown to have a linear response.
Chemical amplifiers are usually only calibrated for one or two types of peroxy radicals.
However, the chain length of each peroxy radical varies, resulting in a different amount of $NO_2$
production depending on the mixture of peroxy radicals present, which could lead to
over/underestimations depending on the ambient mixture. Additionally, the chain length is
significantly affected by humidity due to the increase in $HO_2$ wall loss on wet surfaces and to
an enhanced termination rate of $HO_2$ by reaction with NO to give $HNO_3$. $HO_2$ has been shown
to form a complex with $H_2O$ ($HO_2 \cdot H_2O$), which reacts 4-8 times faster with NO, creating
$HNO_3$, at 50% relative humidity (RH) compared to under dry conditions (Butkovskaya et al.,
2007; Butkovskaya et al., 2009; Duncianu et al., 2020). This leads to the measured chain length
decreasing by a factor of two when going from dry conditions to 40% RH and by a factor of
three at 70% RH (Duncianu et al., 2020; Mihele and Hastie, 1998). Finally, the chain length is
impacted by the gas reagents (NO and CO). Peroxy radicals and alkoxy radicals (RO) can react
with NO to create organic nitrates and nitrites, which terminates the chain reaction, preventing
further radical propagation processes. This is favoured by longer chain peroxy radicals, and at
high NO concentrations. The formation yield of organic nitrates and nitrites differs from a few
percent to up to ~23% depending on the nature of the R group present (Duncianu et al., 2020).



It is therefore important to determine the optimal concentrations of reagent gas for each
individual instrument as it could vary with what material has been used in the reactor.

In the presence of sufficient levels of NO, additional ambient peroxy radicals not

accounted for in photochemical models should lead to an underestimation of the simulated
production rate of $O_3$, which occurs via reactions (4) and (5) followed by photolysis of $NO_2$.
The production of $O_3$ ($P(O_3)$) can be calculated using equation (VI):

$$P(O_3) = k_4[NO][RO_2] + k_5[NO][HO_2] \qquad\qquad (VI)$$

Volz-Thomas et al. (2003) calculated $O_3$ production rates from PSS-derived and

chemical amplification-measured $RO_x$ during the BERLIOZ campaign in Pabstthum,
Germany, resulting in an average of ~ 20 ppbV $h^{-1}$ and ~ 2 ppbV $h^{-1}$ across the campaign,
respectively. The large difference was credited to an unknown process that converts NO into
$NO_2$ without causing additional $O_3$ production (Volz-Thomas et al., 2003). This is possible if
NO is oxidised by an oxidant which also destroys $O_3$, similarly to halogen atoms/halogen
oxides. This hypothesis is consistent with observations by Parrish et al. at a mountain station
in Colorado, where a missing oxidant of photolytic origin was identified (Parrish et al., 1986).
It was shown that if the NO to $NO_2$ oxidation was completely due to $RO_x$, the increased $O_3$
production would result in $O_3$ mixing ratios significantly higher than measured, yet if the
oxidant exhibited similar reaction mechanisms to IO, extremely high (70 pptV) mixing ratios
of IO would be needed (Parrish et al., 1986). These IO levels are more than an order of
magnitude higher than observations in the marine atmosphere (Inamdar et al., 2020; Mahajan
et al., 2010; Prados-Roman et al., 2015; Read et al., 2008).

In regions where the net $O_3$ production is negligible or negative during the day due to

very low NO levels, it is more relevant to compare the net ozone production rate (NOPR) to
the observed change in $O_3$. The chemical NOPR can be calculated as the difference between
the photochemical processes producing and destroying $O_3$:

$$NOPR = P(O_3) - L(O_3) \qquad\qquad (VII)$$

where $P(O_3)$ is determined using equation (VI) and the loss rate of $O_3$ ($L(O_3)$), is usually
determined from reactions (8-12). Additionally, halogens have previously been shown to cause
an $O_3$ loss of $0.23 \pm 0.05$ ppbV $h^{-1}$ in the MBL (initiated by reaction 13)  (Read et al., 2008),
which is in line with other studies suggesting that halogens can have a significant impact on $O_3$
in marine environments (Saiz-Lopez et al., 2012; Sherwen et al., 2016; Vogt et al., 1999).





$O_3 + hv \rightarrow O(^1D) + O_2$      $(\lambda \leq 340 \text{ nm})$      (8)
$O(^1D) + H_2O \rightarrow 2 \text{ OH}$      (9)
$O(^1D) + M \rightarrow O(^3P)$      (10)
$OH + O_3 \rightarrow HO_2 + O_2$      (11)
$HO_2 + O_3 \rightarrow OH + 2 O_2$      (12)
$X + O_3 \rightarrow XO + O_2$      $(X = Br, Cl, I)$      (13)
The actual rate of change of $O_3$ within the planetary boundary layer is also impacted by
the physical processes of advection, deposition and entrainment, which complicates
comparisons with the NOPR. However, if these physical processes change only negligibly over
the course of a day, such as in marine well mixed air masses, their net influence can be deduced
from the net night time change in $O_3$ (Ayers and Galbally, 1995; Ayers et al., 1992; Read et
al., 2008), allowing a calculation of the NOPR from observations. A comparison of the
observed and calculated NOPR gives an indication of whether production and loss rates of $O_3$
from known processes are sufficient to explain the photochemical regime (Read et al., 2008).
From the studies shown in Table 1, there is clearly widespread evidence of enhanced
PSS-derived $RO_2$ compared to measurements and models, however, all methods to derive $RO_x$
are not without challenges as described above. The large uncertainties associated with $RO_x$
measurements, especially at high humidities where the chain length is significantly impacted
by enhanced wall loss and the production of $HNO_3$, suggest that measurements could be
underestimating $RO_x$ in the atmosphere. Previous studies also find that the additional
conversion of NO to $NO_2$ caused by the extra "$RO_2$" should only produce minimal additional
$O_3$, or at least lead to additional $O_3$ destruction, thus inferring an unknown missing oxidant
which exhibits different chemical behaviour to peroxy radicals.
Up to 25% of methane removal occurs in the tropical MBL due to the high
photochemical activity and humidity resulting in high OH radical concentrations (Bloss et al.,
2005). Thus, it is crucially important to understand the fundamental oxidation processes, such
as the $NO_x$-$O_3$ cycle, occurring in this region. However, remote $NO_x$ measurements are rare
due to the difficulty in measuring very low (pptV) mixing ratios. Most previous remote $NO_x$
measurements have taken place during short campaigns and do not give information on
seasonal changes and long-term trends (Carsey et al., 1997; Jacob et al., 1996; Peterson and





Honrath, 1999; Rhoads et al., 1997). Here, we investigate the photostationary state under clean
marine conditions from three years of observations (2017-2020) at the Cape Verde
Atmospheric Observatory (CVAO) in the tropical east Atlantic, representing a unique dataset
to investigate $NO_x$-$O_3$ chemistry in the remote MBL (Andersen et al., 2021; Carpenter et al.,
2010; Lee et al., 2009). We also compare the chemical net $O_3$ production rate (NOPR)
calculated from a box model with NOPR derived from the observed net $O_3$ rate of change, in
order to evaluate the possibility of missing peroxy radicals in this remote environment.

## 3 Methods

### 3.1 Measurements

Year-round measurements of meteorological parameters and trace gases including NO,
$NO_2$, and $C_2$-$C_8$ VOCs have been conducted at the CVAO (16° 51' N, 24° 52' W) since October
2006. The CVAO is located on the north eastern coast of São Vicente, Cabo Verde. The air
sampled predominantly comes from the northeast (see Figure 1) and has travelled over the
Atlantic Ocean for multiple days since the last exposure to anthropogenic emissions, with the
potential exception of ship emissions (Carpenter et al., 2010; Read et al., 2008). This makes it
an ideal location to investigate fundamental photochemistry in an ultra-clean environment.
Wind speed (m/s), wind direction (°), temperature (°C), relative humidity (%),
barometric pressure (mbar) and total solar radiation (W/m$^2$) are measured at a height of 7.5 m
using an automatic weather station from Campbell Scientific. NO and $NO_2$ have been measured
using an ultra-high sensitivity NO chemiluminescence instrument, which measures $NO_2$ by
photolytic conversion to NO, at the CVAO since 2006 (Lee et al., 2009). The technique and
data analysis have been described in detail elsewhere (Andersen et al., 2021). $O_3$ is measured
using a Thermo Scientific 49i Ozone monitor as described in Read et al. (2008). Photolysis
rates of a variety of species were measured in 2020 using a spectral radiometer (a 2-pi sr quartz
diffuser coupled to an Ocean Optics QE65000 spectrometer via a 10 m fibre optic cable). Prior
to 2020, photolysis rates are calculated in this study based on the correlation between the
measured photolysis rates in 2020 and the total solar radiation, as described in the
supplementary information. Average $j$$NO_2$ and $j$$O(^1D)$ for different seasons are shown in Table
2. VOCs are measured using a dual channel Agilent 7890A gas chromatograph coupled with a
Flame Ionization Detector (GC-FID) and a MARKES Thermal Desorption Unit with an ozone





precursor trap that is cooled to -30 °C (Read et al., 2009). Details of the calibration and
uncertainties are given in the World Calibration Centre (WCC)-VOC audit report
(Steinbrecher, 2019). Examples of the VOCs measured at the CVAO can be found in Table 2.
Carbon monoxide (CO), and methane ($CH_4$), are measured using a cavity ring-down
spectrometer (CRDS), G2401 manufactured by Picarro Inc, following the Global Atmosphere
Watch (GAW) recommended technique for long term remote measurements.  The instrument
is highly linear, has a precision of 1 ppbV and 0.3 ppbV over 10 minutes for CO and $CH_4$
respectively and no measurable drift (Zellweger et al., 2016; Zellweger et al., 2012).

Time series of NO, $NO_2$, $O_3$, $jNO_2$, $jO(^1D)$, temperature, CO, propene, benzene and

$CH_4$ for July 2017 – June 2020 are shown in figures S4-S6. The specifics of each instrument
and their respective measurements can be found in Table 2 and a full description of the CVAO
site and associated measurements is given in Carpenter et al. (2010).

### 3.1.1  $NO_2$ Measurement Artefact

One of the drawbacks of measuring $NO_2$ by photolytic conversion to NO is it can be

subject to artefacts. These could either be of a photolytic or thermal origin (Bradshaw et al.,
1999; Gao et al., 1994; Parrish et al., 1990; Ridley et al., 1988; Ryerson et al., 2000). Photolytic
artefacts occur when other compounds containing -NO, $-NO_2$, or $-NO_3$ photolyse to form NO
over a similar wavelength range as $NO_2$ and thereby produce an overestimate of $NO_2$ in the
sample (Pollack et al., 2010). Thermal artefacts are caused by thermally labile compounds
which decompose in photolytic converters when they heat up and release NO that is measured
by the detector or $NO_2$ which is immediately photolytically converted to NO and then detected
(Reed et al., 2016). The maximum potential $NO_2$ artefact can be estimated using measured or
modelled mixing ratios of a range of potential interfering compounds. The photolytic
contribution can be estimated based on the absorption cross section (ACS) of $NO_2$ and the
potential interferents around the peak wavelength of the diodes used to convert $NO_2$ into NO
(385 nm). The ACS of $NO_2$ and some known interfering compounds over the wavelength range
380-390 nm are shown in Table 3. $NO_2$ and most of the interferents, with the exception of
HONO, show relatively invariant ACSs across these wavelengths. When the ACSs of both $NO_2$
and the particular interferent are invariant over the spectral output of the diodes, the ratio at the
peak wavelength is used to estimate the potential artefact. However, since the ACS of HONO
varies significantly over the range, the $HONO/NO_2$ ACS ratio has been estimated assuming a



Gaussian output of the diodes over the wavelengths. It is also important to take into account
whether photolysis of the potential interferent produces $NO_2$ or NO. If $NO_2$ is the product then
it will be photolysed to NO with the same efficiency as $NO_2$ in the ambient air, however, if NO
is the product then 1 converted molecule will be detected as 2 $NO_2$ molecules if the conversion
efficiency of $NO_2$ is 50 %. Organic nitrates, $HNO_3$, and $NO_3$ do not photolyse at 385 nm and
have therefore not been included in the evaluation of photolytic artefacts.

The main potential photolytic artefact for the CVAO $NO_2$ measurements is HONO.

Measurements of HONO at the CVAO using a Long Path Absorption Photometer (LOPAP)
show levels of up to ~ 5 pptV (Reed et al., 2017), indicating an $NO_2$ artefact of up to 0.63 pptV.
However, these measurements were made using a thermostated inlet system with reactive
HONO stripping, where loss of HONO to the sample lines is minimised. The $NO_x$ instrument
at the CVAO samples at the end of the manifold making it highly likely that a significant
fraction of HONO is lost on the manifold before the air is introduced to the $NO_x$ instrument
due to the high surface reactivity of HONO (Pinto et al., 2014). Thus, we regard the potential
HONO-induced artefact of 0.63 pptV as an upper limit. No other potential photolytic artefacts
have been measured at the CVAO, however using the GEOS-Chem model (see section 3.2.2)
we calculated seasonal cycles of 20 potential interfering compounds at the CVAO (Figure S7).
None of these compounds exhibit major seasonal differences, indicating that any measurement
artefact will be fairly constant across the year. The contribution from photolytic degradation of
compounds other than HONO is predicted to be less than 0.05 pptV using the estimated
conversion efficiency of each compound in Table 3 and the modelled mixing ratios at the
CVAO.

Peroxyacetyl nitrate (PAN) is produced in polluted areas and transported to remote

regions, where it can thermally decompose into peroxy radicals and $NO_2$. 5.8% of the available
PAN has been shown to thermally decompose in blue light converters (BLC) switched on 40%
of the time (Reed et al., 2016). This can cause significant overestimations of $NO_2$ in colder
regions where PAN can build up in the atmosphere due to its long lifetime (Kleindienst, 1994),
however, in warmer regions such as Cabo Verde the overestimation will be substantially lower
due to the much shorter lifetime (~ 40-230 minutes at 25°C) (Bridier et al., 1991; Kleindienst,
1994), and hence lower concentration of PAN. At the CVAO, PAN has been measured in
February 2020 using gas chromatography as described by Whalley et al. (Whalley et al., 2004),
however, all measurements were below the limit of detection (LOD) of 6 pptV. The photolytic
converter (PLC) used at the CVAO is only switched on 20% of the time, so a thermal





decomposition efficiency of 5% for PAN is used to estimate a potential artefact of 0.3 pptV
from PAN. Combining photolytic and thermal artefact contributions gives a maximum
potential $NO_2$ artefact of 0.97 pptV at the CVAO, which is within the uncertainty previously
reported for the $NO_2$ measurements, see Table 2 (Andersen et al., 2021).

## 298  3.2  Modelling

### 299  3.2.1  Chemical Box Modelling

A tailored zero-dimensional chemical box model of the lower atmosphere,

incorporating a subset of the Master Chemical Mechanism (MCM v3.3.1) (Jenkin et al., 2015)
into the AtChem2 modelling toolkit (Sommariva et al., 2020), was used to estimate
concentrations of OH, $HO_2$ and $RO_2$ and daily chemical production and loss of $O_3$ at the CVAO.
The MCM describes the detailed atmospheric chemical degradation of 143 VOCs, through
17,500 reactions of 6900 species. More details can be found on the MCM website
(http://mcm.york.ac.uk, last access: 4th March 2022). A fixed deposition rate of 1.2 x $10^{-5}$ $s^{-1}$
was applied to all model generated species, giving them a lifetime of approximately 24 hours.
The model was constrained to 34 observationally derived photolysis rates, temperature,
pressure, and relative humidity, along with a range of observed chemical species, defined in
Table 2.

### 312  3.2.2  GEOS-Chem

Concentrations of 20 different chemical species were extracted every hour during 2019

at nearest point in space and time from the GEOS-Chem model (v12.9.0,
DOI:10.5281/zenodo.3950327). The v12.9.0 model as described by Wang et al. (2021) was run
at a nested horizontal resolution of 0.25x0.3125 degrees over the region (-32.0 to 15.0 °E, 0.0
to 34.0 °N), with boundary conditions provided by a separate global model run spun up for one
year and with acid uptake on dust considered as described by Fairlie et al. (2010) (Fairlie et al.,
2010; Wang et al., 2021).



## 4  Results and Discussion


Monthly diurnal cycles of $HO_2$, $RO_2$, and OH were modelled by constraining the box
model to the measurements described in Table 2 (except $NO_2$) using hourly median
concentrations for each month from July 2017 – June 2020 where all the trace gas
measurements were available. When measured $jO(^1D)$ was not available, the hourly average
from the same month across the other years was used. Calculated photolysis rates based on
total solar radiation (see supplementary) were used up to December 2019 for all other
photolysis rates than $jO(^1D)$.
The modelled OH, $HO_2$ and $RO_2$ concentrations agree reasonably well with previous
measurements from short term field campaigns based at the CVAO and from various cruises
in the Atlantic Ocean (see Figure 2). All the previous measurements of $RO_x$ ($HO_2$ + $RO_2$)
shown in Figure 2 were conducted using the chemical amplifier technique, which is subject to
high uncertainties due to the challenges described above. Daily diurnal cycles of $RO_2$ and $HO_2$
for 9 days in August 2017, 12 days in October 2017, and 20 days in January 2018 were
modelled to investigate their daily variability (see Figure S8). Seasonal differences can be
observed from the daily outputs, but no major day to day changes within a given month.

## 4.1  Comparison of measured and PSS $NO_2$ concentrations

Daily midday (12.00-15.00 UTC, local+1) $NO_2$ mixing ratios were calculated from the
Leighton ratio using equation II ($[NO_2]_{PSS}$), the measured NO, $O_3$, and $jNO_2$ and $k_1 = 2.07 \times$
$10^{-12} \times e^{(-1400/T)}$ (Atkinson et al., 2004) for a three-year period (July 2017 – June 2020). Figure
3A shows that $[NO_2]_{PSS}$ significantly underestimates the measured $NO_2$, indicating that
additional oxidants are needed to convert NO into $NO_2$. Daily midday values of $[NO_2]_{PSS\ ext.}$
were calculated using equation III, where a midday average of each modelled monthly diurnal
cycle of $HO_2$ and $RO_2$ in Figure 2 was used for all days of their respective month together with
previous yearly averaged midday measurements of IO ($1.4 \pm 0.8$ pptV) and BrO ($2.5 \pm 1.1$
pptV) (Mahajan et al., 2010; Read et al., 2008) at the CVAO. $RO_2$ was assumed to be equivalent
to $CH_3O_2$, making $k_4 = 2.3 \times 10^{-12} \times e^{(360/T)}$, $k_5 = 3.45 \times 10^{-12} \times e^{(270/T)}$, $k_6 = 7.15 \times 10^{-12} \times e^{(300/T)}$,
and $k_7 = 8.7 \times 10^{-12} \times e^{(260/T)}$ (Atkinson et al., 2004). $[NO_2]_{PSS\ ext.}$ was calculated using a midday
average of the modelled monthly $[HO_2]$ and $[RO_2]$ in Figure 2 as well as the modelled daily
midday averages from the diurnal cycles in Figure S8 for August 2017, October 2017, and





January 2018. A scatter plot of monthly vs daily calculated $[NO_2]_{PSS\ ext.}$ around the 1:1 line (see
Figure S9) verifies the use of monthly calculated $[HO_2]$ and $[RO_2]$ for the remaining analyses.

Figure 3B shows that the agreement between measured $NO_2$ and $[NO_2]_{PSS\ ext.}$ was

improved significantly by including modelled additional oxidants. At $NO_2$ mixing ratios below
20 pptV, the scatter of $[NO_2]_{PSS\ ext.}$ vs $[NO_2]_{Obs.}$ was close to the 1:1 line, however, at higher
$NO_2$ mixing ratios $[NO_2]_{PSS\ ext.}$ under-predicts the observed $NO_2$ mixing ratio by on average
9.5 pptV. $NO_2$ mixing ratios above 20 pptV are predominantly observed at the CVAO from
December-February (Andersen et al., 2021), which coincides with the arrival of predominantly
African air to the site (see Figure 1).

We next investigate the effects of seasons and the abundance of NO on the ability of

the full PSS equation (equation III) to predict $NO_2$. Daily midday averages of
$[NO_2]_{Obs.}/[NO_2]_{PSS\ ext.}$ are plotted as a function of NO in Figure 4. A ratio of 1 would be
expected if all relevant reaction mechanisms have been taken into account. The deviations from
1 in the ratio can be observed to increase with decreasing NO mixing ratio during March-
December. The dashed lines in Figure 4 visualise the effect of a constant $NO_2$ artefact of 0.97
pptV (our calculated upper limit) on the $[NO_2]_{Obs.}/[NO_2]_{PSS\ ext.}$ ratio, showing that the artefact,
while small, can explain some of this observed trend. However, only a small dependence on
the NO mixing ratio is seen for January and February, where enhancements of
$[NO_2]_{Obs.}/[NO_2]_{PSS\ ext.}$ above 1 continue out to 10 pptV of NO.  At Hohenpeissenberg, Germany,
similar trends with increasing $NO_2/NO$ ratio with decreasing NO have been observed, which
were partly explained by measurement uncertainty in NO and partly by the PSS not being
established after being perturbed by $NO_x$ emissions or variable $jNO_2$ (Mannschreck et al.,
2004). An opposite trend to that observed here and at Hohenpeissenberg was observed over the
South Atlantic Ocean, with increasing deviations in $[NO_2]_{Obs.}/[NO_2]_{PSS\ ext.}$ with increasing $NO_2$
from 3-20 pptV (Hosaynali Beygi et al., 2011), which was explained by a missing photolytic
oxidation process.

## 4.2   $NO_2$ Artefact or Missing Oxidant?

Deviations between $[NO_2]_{Obs.}$ and $[NO_2]_{PSS\ ext.}$ are usually attributed to an unaccounted

artefact in the $NO_2$ measurements or a missing oxidant converting NO into $NO_2$ (Bradshaw et
al., 1999; Carpenter et al., 1998; Crawford et al., 1996; Hauglustaine et al., 1999; Hauglustaine





et al., 1996; Hosaynali Beygi et al., 2011; Volz-Thomas et al., 2003). As discussed above, we
show that below 5 pptV of ambient NO, our calculated maximum $NO_2$ artefact of 0.97 pptV
starts to have an impact on the $[NO_2]_{Obs.}/[NO_2]_{PSS\ ext.}$ ratio, however, it is not enough to explain
the enhancements observed, especially in wintertime at the CVAO.
The production of $RO_2$ and $HO_2$ radicals is dependent on the abundance of their VOC
and CO precursors as well as on photochemical activity. To investigate whether the availability
of VOCs, CO or sunlight was related to the discrepancy between $[NO_2]_{Obs.}$ and $[NO_2]_{PSS\ ext.,}$
Figure 4 was replotted by colouring the $[NO_2]_{Obs.}/[NO_2]_{PSS\ ext.}$ ratio as a function of the mixing
ratio of a particular precursor or $jNO_2$ (Figure 5). The high deviations in $[NO_2]_{Obs.}/[NO_2]_{PSS\ ext.}$
at NO > 2.5 pptV can be observed to be associated with higher measured mixing ratios of CO,
ethane, and acetylene, and lower midday $jNO_2$, however, it should be noted that the variation
in midday photolysis rates at the CVAO over the year is relatively small. At similar $jNO_2$ as
observed at the CVAO at midday (>0.007 s$^{-1}$), Hosaynali Beygi et al. observed the largest
deviations in $[NO_2]_{Obs.}/[NO_2]_{PSS\ ext.}$ (Hosaynali Beygi et al., 2011). For the high enhancements
in $[NO_2]_{Obs.}/[NO_2]_{PSS\ ext.}$ at NO < 2.5 pptV at the CVAO, the trends are not as clear. The mixing
ratios of CO can be observed to remain enhanced, however, high $jNO_2$ is seen at NO < 2.5
pptV while the ethane and acetylene mixing ratios are lower than when NO > 2.5 pptV. It is
important to note though that the deviation at very low NO can on most days be explained by
the measurement uncertainty in NO (~1.4 pptV). Figure 5 shows that the abundances of ethene
and propene, both of which have atmospheric lifetimes of less than 3 days, do not seem to
affect the deviation of $[NO_2]_{Obs.}/[NO_2]_{PSS\ ext.}$ from 1. Conversely, high abundances of CO,
ethane, and acetylene, which all have atmospheric lifetimes above 6 weeks (Atkinson et al.,
2006), are observed to be associated with higher $[NO_2]_{Obs.}/[NO_2]_{PSS\ ext.}$ ratios. This could
indicate that long-range transport of pollutants supplies additional peroxy radicals (or other NO
to $NO_2$ oxidants) at the CVAO, which are not predicted from known sources and
photochemistry.
To further evaluate the impact of pollution, $[NO_2]_{Obs.}/[NO_2]_{PSS\ ext.}$ was separated into
three categories based on CO mixing ratios; CO < 90 ppbV, 90 ppbV < CO < 100 ppbV, and
CO > 100 ppbV. The deviations of $[NO_2]_{Obs.}/[NO_2]_{PSS\ ext.}$ from 1 increase with increasing [CO],
with 50$^{th}$ (25$^{th}$-75$^{th}$) percentiles of 1.10 (0.82 -1.37) for CO < 90 ppbV, 1.20 (0.97-1.54) for 90
ppbV < CO < 100 ppbV, and 1.50 (1.18-1.78) for CO > 100 ppbV.  The small deviation from
1, which is within the uncertainty of our measurements (see below), for CO < 90 ppbV is strong
evidence that fundamental oxidation process in ultra-clean marine air, where the main


precursors of $RO_2$ and $HO_2$ are $CH_4$ and CO giving $CH_3O_2$ and $HO_2$, respectively, are well
understood.
An $NO_2$ artefact of 0.7 pptV would reduce the ratio of 1.10 to 1.00 in air masses with
CO < 90 ppbV. Since the minimum value of the artefact is 0 pptV (if there was no conversion
of interferent compounds to NO or $NO_2$), and our estimated upper limit is 0.97 pptV, we
therefore consider it a reasonable assumption that the average $NO_2$ artefact of our instrument
at the CVAO is 0.7 pptV. We make the simple *a priori* assumption that this applies across all
measurements during the period of analyses. Such an artefact is insignificant when considering
total $NO_x$ concentrations, however, it has a non-negligible impact when investigating $NO_2$/NO
ratios in this very low $NO_x$ environment.
Subtracting 0.7 pptV from all the $NO_2$ observations results in median ($25^{th}$-$75^{th}$
percentiles) ratios of 1.00 (0.76-1.29) for CO < 90 ppbV, 1.14 (0.89-1.47) for 90 ppbV < CO
< 100 ppbV, and 1.42 (1.12-1.68) for CO > 100 ppbV (Table 4). Distributions of each category
are plotted in Figure 6A. When CO is between 90 and 100 ppbV, the distribution of
$[NO_2]_{Obs.}/[NO_2]_{PSS\ ext.}$ shows the highest occurrences at ratios of ~1 and ~1.5. When CO > 100
ppbV, it is evident that either additional oxidants are needed to convert NO to $NO_2$, or an
additional $NO_2$ artefact of the order of 4.4 pptV is present in these air masses. As an artefact of
0.7 pptV has already been subtracted, and measurements of HONO and PAN and modelled
mixing ratios of halogen nitrates indicate a fairly stable artefact across the year, 4.4 pptV of
additional artefact seems highly unlikely. This leaves the possibility of a missing oxidant when
the sampled air is enhanced in CO.
Using equation (IV) and (V), the required $RO_x$ ($RO_2$ + $HO_2$) and XO (IO + BrO)
concentrations needed to reconcile $[NO_2]_{Obs.}$ with $[NO_2]_{PSS\ ext}$ can be estimated using $k_{4,5}$ = 2.3
$\times 10^{-12} \times e^{(360/T)}$ and $k_{6,7} = 8.7 \times 10^{-12} \times e^{(260/T)}$ (Atkinson et al., 2004). Our calculations are based
on two scenarios: (1) that the measured [BrO] and [IO] are correct and there is missing $RO_x$,
or (2) that the modelled $[RO_x]$ is correct and there is missing [XO]. Due to the similar rate
coefficients for IO and BrO reacting with NO, a combined XO can be estimated. The results
are summarised in Table 4 based on the three CO categories. The median required $RO_x$ was
determined to be 65.0 (33.68 - 112.5, $25^{th}$-$75^{th}$ percentile) pptV and 109.7 (63.14 - 149.5, $25^{th}$-
$75^{th}$ percentile) pptV for 90 ppbV < CO < 100 ppbV and CO > 100 ppbV, respectively. $RO_x$
measurements during the ALBATROSS cruise varied from 40-80 pptV while in the North
Atlantic, however, with a reported uncertainty of 25% (1σ) they could be as high as 100 pptV





(Burkert et al., 2001). Such concentrations are comparable to the required median $RO_x$ in this
study of 109.7 pptV when $CO > 100$ ppbV. The uncertainty reported for ALBATROSS is
similar to many other studies which have reported 10-36% uncertainty on chemical
amplification $RO_x$ measurements (Cantrell et al., 1997; Clemitshaw et al., 1997; Handisides et
al., 2003; Hernández et al., 2001; Hosaynali Beygi et al., 2011; Volz-Thomas et al., 2003),
however, a recent study in the Pearl River Delta reported an uncertainty of 60% ($1\sigma$) (Ma et
al., 2017). This combined with measurements up to ~150 pptV of $RO_x$ in the South Atlantic
Ocean (Hosaynali Beygi et al., 2011) indicates that our required $RO_x$ levels of ~ 100 pptV may
not be unrealistic in the MBL.
The median required $RO_x$ ($[RO_x]_{PSS}$) can be observed to be ~2.5 times higher than those
modelled for air masses where $CO > 100$ ppbV, whereas the required [XO] is a factor of ~6.5
higher than previous observations at the CVAO (Mahajan et al., 2010; Read et al., 2008) due
to the lower rate coefficients for halogen oxides with NO. Across the three categories, the daily
median ratio of $[RO_x]_{PSS}/[RO_x]_{Model}$ is 1.5, which is similar to those observed in previous
studies both in remote and rural regions (see Table 1). The missing XO required to reconcile
$[NO_2]_{Obs.}$ with $[NO_2]_{PSS\ ext.}$ was determined for each CO category by subtracting the previous
measured average concentration of 3.9 pptV (2.5 pptV BrO + 1.4 pptV IO) from the required
XO. Since CO, the main precursor for $HO_2$, is constrained by measurements in the model, the
calculated $[HO_2]$ is assumed to be correct. Thus, we estimate the required and missing $RO_2$
assuming it is all in the form of $CH_3O_2$ from:
$$[RO_2]_{Required} = \frac{jNO_2[NO_2] - (k_1[O_3] + k_5[HO_2] + k_6[IO] + k_7[BrO])[NO]}{k_4[NO]} \qquad \text{(VIII)}$$

$$[RO_2]_{Missing} = \frac{jNO_2[NO_2] - (k_1[O_3] + k_5[HO_2] + k_6[IO] + k_7[BrO])[NO]}{k_4[NO]} - [RO_2]_{model} \qquad \text{(IX)}$$

Figures 6B and C, show that the missing $RO_2$ or XO level increases with increasing
[CO], reaching a median of 61.3 pptV and 22.7 pptV, respectively, for air masses where $CO >$
100 ppbV, which is approximately 2.2 times the modelled $RO_2$ and 5.5 times the measured XO
in the same air masses. Such an increase in peroxy radicals would, under more polluted
conditions, cause a major increase in $O_3$ production during a day (Volz-Thomas et al., 2003).
We next examine the impact of missing $RO_2$ on net $O_3$ production in Cabo Verde.





## 4.3 Chemical O₃ Loss

The daily chemical loss of $O_3$ between 09.30 (09.00-10.00) and 17.30 (17.00-18.00) UTC was used to evaluate whether the PSS-derived $[RO_2]$ was consistent with the net chemical destruction of $O_3$ at the CVAO. As discussed above, the measured $O_3$ mixing ratio in the MBL is affected by loss mechanisms in the form of photolysis, reactions with $HO_x$ and halogens, and deposition, and by production through $NO_2$ photolysis and by entrainment from the $O_3$-enriched free troposphere. Due to the very stable meteorological condition of the MBL, the variability in entrainment and deposition between night and day is expected to be negligible (Ayers and Galbally, 1995; Ayers et al., 1992; Read et al., 2008). A combined entrainment/deposition term can therefore be estimated from night time $O_3$ measurements, when there is no photochemical production or loss. An hourly entrainment/deposition term was determined for each month using the average change in $O_3$ between 22.30 (22.00-23.00) and 03.30 (03.00-04.00), and found to vary from 0.18 ppbV h$^{-1}$ in January to 0.35 ppbV h$^{-1}$ in May, which is in good agreement with previous measurements at the CVAO of 0.18-0.48 ppbV h$^{-1}$ (Read et al., 2008). The observed daily change in $O_3$ ($\Delta O_{3\ obs.}$) (09.30-17.30) was determined to be $-0.40 \pm 0.32$ ppbV h$^{-1}$ (1σ) across the three years (2017-2020), which is almost identical to the $-0.41 \pm 0.33$ ppbV h$^{-1}$ (1σ) observed at the CVAO in 2007 (Read et al., 2008), but roughly 2 times the daily $\Delta O_{3\ obs.}$ in baseline air at Cape Grim ($-0.24 \pm 0.32$ ppbV h$^{-1}$, 1σ) and Mace Head ($-0.20 \pm 0.21$ ppbV h$^{-1}$, 1σ) (Carpenter et al., 1997) and 2-40 times the modelled $O_3$ loss at Mauna Loa ($-0.01$ to $-0.21$ ppbV h$^{-1}$) (Cantrell et al., 1996; Ridley et al., 1992).

By subtracting the monthly average entrainment/deposition term from the observed daily $\Delta O_3$, the daily chemical loss of $O_3$, $\Delta O_{3\ chem.}$, is obtained. The observations were filtered to exclude periods where the change in CO concentration over the interval period, $\Delta CO$, was outside 1 standard deviation of the mean $\Delta CO$, to avoid the $\Delta O_3$ determination being affected by changing air masses. The resulting observed chemical loss of $O_3$ is averaged by month and plotted in black in Figure 7. $\Delta O_{3\ chem.}$ can be observed to follow photochemical activity, with the lowest $\Delta O_{3\ chem.}$ in October-February, where the lowest photolysis rates are measured (see supplementary and Table 2) and highest $\Delta O_{3\ chem.}$ in March-May and September. A small decrease in $\Delta O_{3\ chem.}$ in June-August occurred simultaneously to the small drop in photolysis rates in June-August. Overall, $\Delta O_{3\ chem.}$ varied from $-0.48$ ppbV h$^{-1}$ in January to $-0.88$ ppbV h$^{-1}$ in May.





In order to evaluate whether these observationally-derived chemical loss rates of $O_3$ are
consistent with PSS-derived peroxy radical concentrations, $\Delta O_3$ chem. was estimated using a
chemical box model incorporating the MCM, as described in section 3.2.1. The model was
constrained to all the measurements described in Table 2, except $NO_2$ and $O_3$, which were left
unconstrained. $\Delta O_3$ chem. was simulated with modelled $[RO_2]$ and $[HO_2]$, with (blue line in
Figure 7) and without (grey in Figure 7) inclusion of the halogen chemistry described in Table
S1, allowing an evaluation of the $O_3$ loss due to halogens, as previously discussed by Read et
al. (2008). Simulations were also performed with $[CH_3O_2]$ constrained to the required $RO_2$,
modelled $[HO_2]$ and including halogen chemistry (orange in Figure 7). In model runs with
halogen chemistry, BrO and IO were constrained to previously measured annual averages ±
reported uncertainties (blue shaded area in Figure 7) (Read et al., 2008). Diurnal cycles of the
required $RO_2$ were constructed using the median of the daily midday averages for each month
determined using equation (VIII) for the peak concentration at midday, 1 pptV overnight and
interpolating linearly in between.
Figure 7 shows that all three modelled $\Delta O_3$ chem. exhibited very similar seasonality as
the observed $\Delta O_3$ chem.. The difference between running the model with and without halogen
chemistry was $0.24 \pm 0.02$ ppbV h$^{-1}$ (1σ), which is almost equivalent to the results of Read et
al. (2008) from the CVAO of $0.23 \pm 0.05$ ppbV h$^{-1}$ (1σ). From May-December, the modelled
$\Delta O_3$ chem. was almost identical whether using modelled $RO_2$ or constraining $CH_3O_2$ to the
required $RO_2$, and both were very similar to observed $\Delta O_3$ chem.. The largest difference in $\Delta O_3$
chem. between using modelled $RO_2$ and constraining $CH_3O_2$ is observed in January where the
difference reached 0.09 ppbV h$^{-1}$, however, this is caused by constraining $CH_3O_2$ to 100 pptV,
which is 5 times more than the modelled $RO_2$. The average difference between the observed
and modelled $\Delta O_3$ chem. is $0.06 \pm 0.07$ ppbV h$^{-1}$ (1σ) when constraining $CH_3O_2$ to the required
$RO_2$ and $0.04 \pm 0.07$ ppbV h$^{-1}$ (1σ) when using modelled $RO_2$.
Overall, the very small differences in modelled $\Delta O_3$ chem. whether including the
"missing $RO_2$" or not are a function of the $NO_x$-limited conditions of the remote MBL, where
$O_3$ production is relatively insensitive to the mixture and abundance of peroxy radicals
(Sillman, 1999). Thus, although our analysis shows that peroxy radicals with the equivalent
$O_3$ production potential as $CH_3O_2$ cannot be ruled out as the missing oxidant in marine air
masses with aged pollution, neither does it provide robust evidence that the missing oxidant is
$O_3$-producing. Nevertheless, the deviation between PSS-derived peroxy radicals in this study



and previous measurements can potentially be explained by the difficulty in measuring peroxy
radicals, as discussed above.

## 5    Conclusions

In the remote MBL (CO < 90 ppbV, $NO_x$ < 43 pptV ($90^{th}$ percentile = 23 pptV)) we
have shown that the observed $NO_2/NO$ ratio is consistent with fundamental photochemical
theory, and that neither missing oxidants nor deviations of the photostationary state are required
to reconcile observations with the calculated $NO_2/NO$ ratio. This is to our knowledge the first
time this has been shown in a low $NO_x$ environment. However, observed $NO_2$ levels became
increasingly higher than predicted as the CO mixing ratio increased and the air more influenced
by long range transport of air pollution in winter. A detailed analysis of potential $NO_2$
measurement artefacts at the CVAO showed that such artefacts were unlikely to account for
these deviations, thus we evaluated the case for a missing NO to $NO_2$ oxidant. The required
oxidant in air masses with CO > 100 ppbV reached a median of 109.7 pptV when treated as
$CH_3O_2$. These levels are ~ 2.5 times higher than both our modelled $RO_x$ ($RO_2$ + $HO_2$) and
previous measurements of $RO_x$ measured by chemical amplification at the CVAO. However,
chemical amplification measurements are known to be highly uncertain due to the difficulty in
determining the chain length of the mixture of $RO_2$ in the ambient matrix, and we note that the
modelled $O_3$ production at the CVAO, with the inclusion of these additional peroxy radicals,
did not deviate significantly from the observed $O_3$ production.  Overall, we conclude that there
is strong evidence for a missing oxidant in remote marine air impacted by long range transport
of pollution, and that peroxy radicals cannot be ruled out as to their identity.

## 6    Acknowledgements

The authors would like to thank the UK Natural Environment Research Council/
National Centre for Atmospheric Science (NERC/NCAS) through the Atmospheric
Measurement and Observation Facility (AMOF) for funding the CVAO programme. STA's
PhD was supported by the SPHERES Natural Environment Research Council (NERC)
Doctoral Training Partnership (DTP), under grant NE/L002574/1. LJC acknowledges funding



from the European Research Council (ERC) under the European Union's Horizon 2020 pro-
gramme (project O3-SML; grant agreement no. 833290).

## 7   Author Contributions

Data analysis has been performed by STA. The box model has been run by BSN. Back

trajectories have been modelled by MR. GEOS-Chem has been run by TS. The instruments at
the CVAO have been run by STA, KAR, SP, JH, and LN. KAR and LKW have processed the
spectral radiometer data. The manuscript has been written by STA, LJC, JDL, BSN, and KAR.

## 8   Additional Information

The authors declare that they have no competing interests.


## 8.1   Data availability:

$NO_x$, VOCs, meteorological data, CO and $O_3$: WDCRG (World Data Centre for

Reactive Gases)/Norwegian Institute for Air Research (NILU) EBAS database (EBAS
(nilu.no))

$CH_4$ and CO: WDCGG (World Data Centre for Greenhouse Gases) (kishou.go.jp)





# 9  Figures

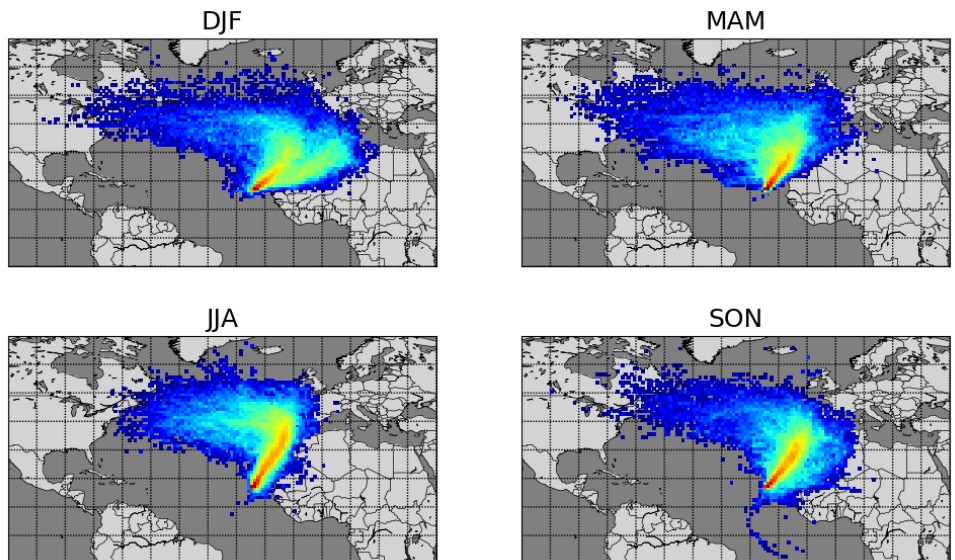


Figure 1: Seasonal average 10-day back trajectories for the CVAO determined using
FLEXPART as described in Andersen et al. (2021).

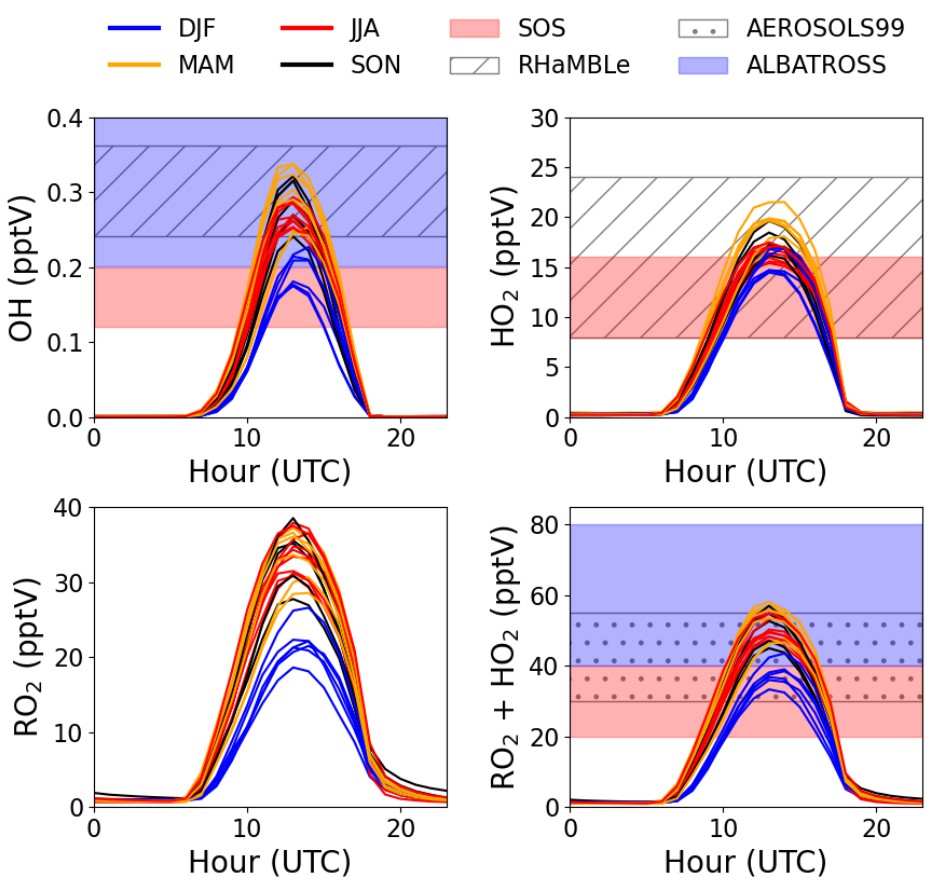


Figure 2: Average monthly diurnal cycles of modelled OH, HO₂, RO₂, and HO₂+RO₂ coloured by season compared to midday measurements during SOS (Carpenter et al., 2010; Vaughan et al., 2012), RHaMBLe (Whalley et al., 2010), AEROSOLS99 (Hernández et al., 2001), and ALBATROSS (Burkert et al., 2001).



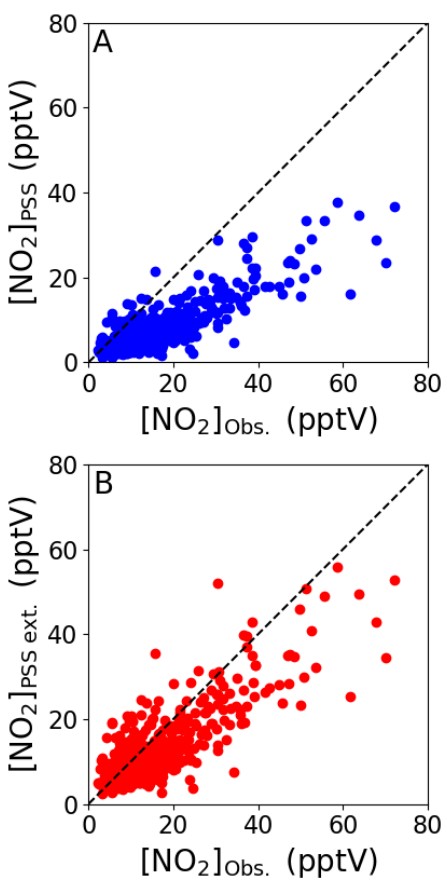

Figure 3: Midday (12.00-15.00 UTC, local+1) daily averages of $[NO_2]_{PSS}$ (A) and $[NO_2]_{PSS\ ext.}$ (B) plotted against the observed $NO_2$ using measurements from July 2017 – June 2020. The black dashed lines show the 1:1 ratio.



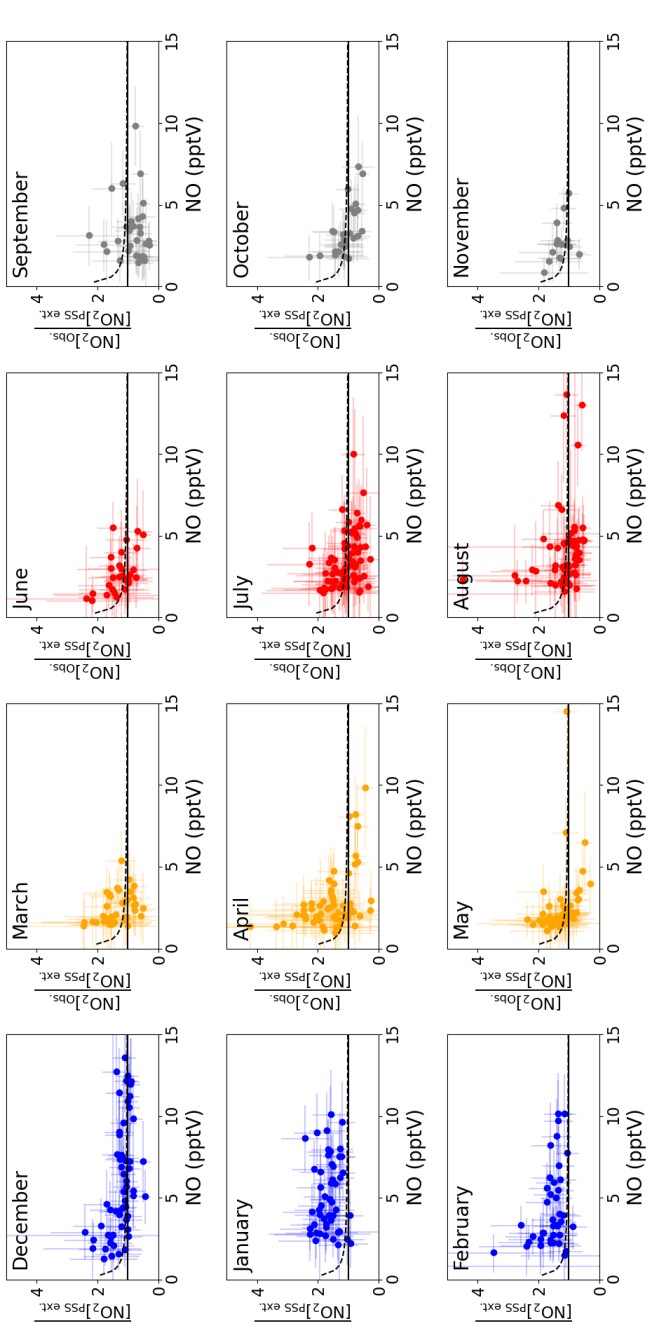

Figure 4: Monthly plots of midday (12.00-15.00 UTC, local+1) daily averages of [NO$_2$]$_{Obs.}$/[NO$_2$]$_{PSS\ ext.}$ vs. the measured NO mixing ratio. The solid lines represent a ratio of 1 between the observed and predicted NO$_2$. The error bars represent ± 2σ on the calculated ratio and measured NO. The dashed lines represent ([NO$_2$]$_{PSS\ ext.}$ + 0.97 pptV)/[NO$_2$]$_{PSS\ ext.}$ to visualise the effect of a NO$_2$ artefact of 0.97 pptV on the ratio using the average measured $j$NO$_2$ and O$_3$ and modelled HO$_2$ and RO$_2$ for each month and the annually average measured IO and BrO for the CVAO. The uncertainty of each data point has been determined from measurement uncertainties in Table 2, the uncertainties in the measured BrO and IO described in the text, and 20% uncertainty on all the rate coefficients. The uncertainty in the modelled radicals has not been included


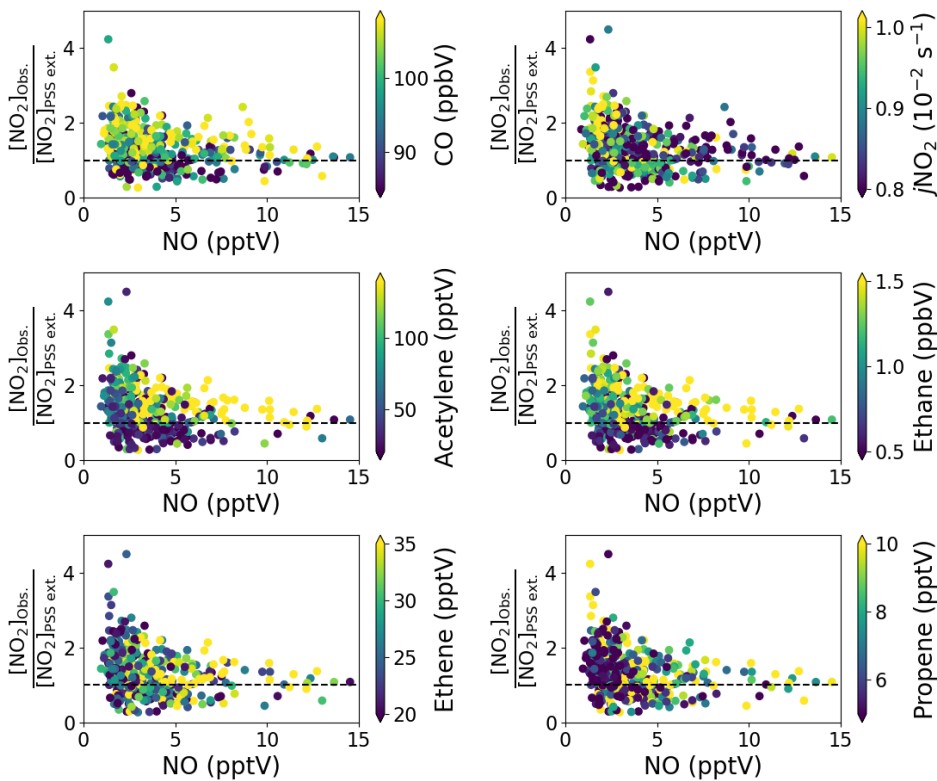


Figure 5: Midday (12.00-15.00 UTC, local +1) daily averages of $[NO_2]_{Obs.}/[NO_2]_{PSS\ ext.}$ from July 2017 to June 2020 plotted against the measured NO and coloured by five different measured precursors for either $HO_2$ or $RO_2$ and $jNO_2$. The dashed line represent a ratio of 1






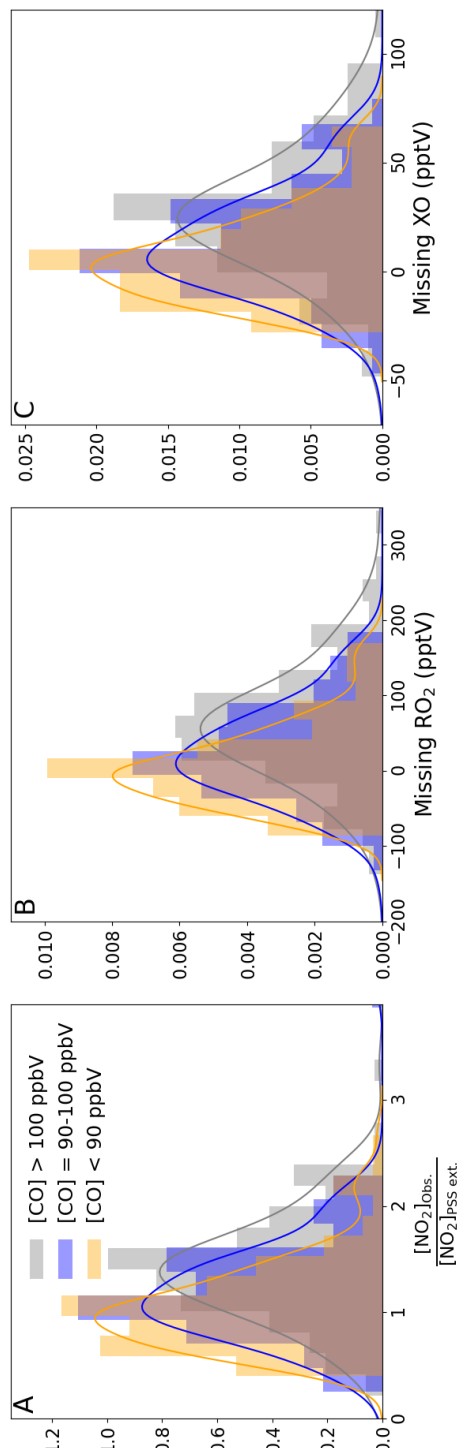

Figure 6: Density distributions of (A) $[NO_2]_{Obs.}/[NO_2]_{PSS\ ext.}$, (B) missing $RO_2$, and (C) missing XO separated by measured CO mixing ratios. An $NO_2$ artefact of 0.7 pptV has been subtracted from all data.



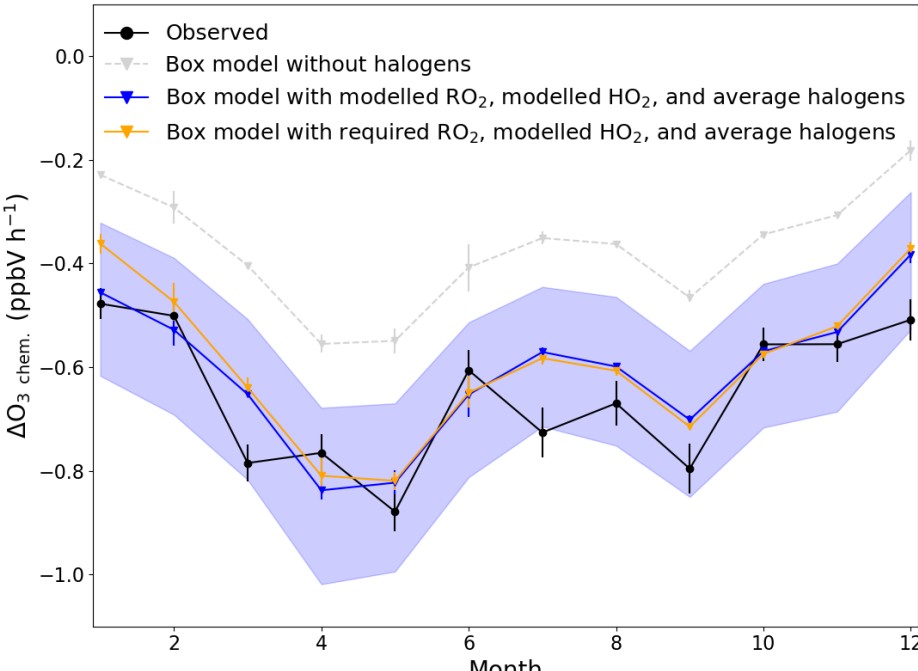


Figure 7: Average monthly $\Delta O_3$ due to chemical loss between 09.30 (09.00-10.00) and 17.30 (17.00-18.00) UTC for each month (black) compared to box modelled $\Delta O_3$ due to chemical loss using modelled $RO_2$ and $HO_2$ with (blue) and without (grey) halogen monoxides (BrO and IO), and using required $RO_2$ to get $[NO_2]_{Obs.}/[NO_2]_{PSS\ ext.} = 1$, modelled $HO_2$, and the annually averaged halogen monoxides (orange). The error bars on the observed chemical loss is the standard error of all the days used for each month and for the box model it is the minimum and maximum $\Delta O_3$ modelled for each month. The blue shaded area show the possible variability in the chemical loss when including the measured halogens at the CVAO (BrO; $2.5 \pm 1.1$ pptV, IO; $1.4 \pm 0.8$ pptV) (Read et al., 2008).




10 Tables


**Table 1: Summary of previous studies which have compared [RO$_x$]$_{PSS}$ against measured and/or modelled [RO$_x$] in rural, marine and remote conditions.**

| Location | NO$_x$ instrument | NO$_x$ | φ$^a$ | $\frac{[RO_x]_{PSS}}{[RO_x]_{Measured}}$ $^b$ | $\frac{[RO_x]_{PSS}}{[RO_x]_{Model}}$ $^b$ | $\frac{[RO_x]_{Measured}}{[RO_x]_{Model}}$ $^b$ | Reference |
|---|---|---|---|---|---|---|---|
| **Rural conditions** | | | | | | | |
| Hohenpeissenberg, Germany | CLD with PLC$^c$ | NO; 50-7000 pptV | 2-5.7$^d$ | 2-3$^e$ | - | - | (Mannschreck et al., 2004) |
| Pearl River Delta, China | CLD with PLC$^c$ | NO; 50-4000 pptV | 1-8.5$^d$ | ~1$^e$ | 2-10 | ~2$^e$ | (Ma et al., 2017) |
| Pabstthum, Germany | CLD with PLC$^c$ | 1-7 ppbV | 1.1-3.0$^d$ | ~4$^e$ | - | - | (Volz-Thomas et al., 2003) |
| Idaho Hill, Colorado | CLD with PLC$^c$ | 38 pptV-21.3 ppbV | - | 2.1 (mean)$^e$ | - | ~1$^{e,f}$ | (Cantrell et al., 1997; Williams et al., 1997) |
| Pine forest, Alabama | CLD with PLC$^c$ | 1-5 ppbV | - | 1-2$^e$ | - | ~1$^{e,f}$ | (Cantrell et al., 1992; Cantrell et al., 1993a; Parrish et al., 1986) |
| Essex, England | CLD with Mo$^g$ | NO; 0.3-9.9 ppbV | - | - | - | ~1.4$^e$ | (Emmerson et al., 2007) |
| Ponderosa pine forest, Rocky Mountains | CLD with PLC$^c$ | NO; 100-150 pptV | - | - | - | <3$^h$ | (Wolfe et al., 2014) |
| **Marine/Remote with pollution** | | | | | | | |
| Arabian Peninsula | CLD with PLC$^c$ and CRDS$^i$ | < 50 pptV - > 10 ppbV | - | - | ~ 1 | - | (Tadic et al., 2020) |
| Amazon Basin (Manau) | CLD with PLC$^c$ | 100 pptV - 30 ppbV | 1-6$^d$ | - | ~ 1$^k$ | - | (Trebs et al., 2012) |
| **Marine/Remote conditions** | | | | | | | |
| South Atlantic Ocean | CLD with PLC$^c$ | NO$_2$; 3-20 pptV | 1-12.5$^l$ | 1.27$^e$ | ~ 5 | ~ 4$^e$ | (Hosaynali Beygi et al., 2011) |
| Mauna Loa, Hawaii | CLD with PLC$^c$ | 20-60 pptV | 1.4-2.2 | 1.5-3$^e$ | 2-3.5 | 1.2-2$^e$ | (Hauglustaine et al., 1996) |
| Mace Head, Ireland | CLD with TC$^m$ | NO < 10 pptV | - | - | - | ~ 0.25$^e$ | (Carpenter et al., 1997; Cox, 1999) |
| Cape Grim, Tasmania | CLD with PLC$^c$ | NO < 5 pptV | - | - | - | ~ 0.4$^e$ | (Carpenter et al., 1997; Cox, 1999) |
| Cabo Verde | CLD with PLC$^c$ | <50 pptV | 0.45-12.0$^d$ (median = 2.1) | - | 1.5 (median) | - | This study |




$^a$Without radicals and halogens. $^b[RO_x] = [HO_2] + [RO_2]$. $^c$CLD with PLC = Detection by
chemiluminescence with photolytic converter for $NO_2$. $^d$Increasing φ with decreasing [NO],
$[NO_2]$ or $[NO_x]$. $^e[RO_x]$ measured by chemical amplification. $^f$Calculated/modelled using stead
state theory. $^g$CLD with Mo = Detection by chemiluminescence with molybdenum converter.
$^h[RO_x]$ measured by Peroxy Radical Chemical Ionization Mass Spectrometry (PeRCIMS).
$^i$CRDS = Cavity Ring down spectroscopy. $^k$PSS derived $[RO_x]$ was within the range of the
modelled values. $^l$Increasing φ with increasing $[NO_2]$. $^m$CLD with TC = Detection by
chemiluminescence with thermal converter.






**Table 2: Overview of instruments and measurements used from the CVAO.**

| Instrument | Measurement | Accuracy | DJF[a] | MAM[a] | JJA[a] | SON[a] | Reference[b] |
|---|---|---|---|---|---|---|---|
| AQD | NO (pptV) | 1.4 pptV | 5.3 ± 7.8 | 1.9 ± 4.2 | 2.7 ± 5.6 | 3.6 ± 5.9 | Andersen et al. (2021) |
| | NO$_2$ (pptV) | 4.4 pptV | 27.0 ± 35.8 | 10.0 ± 13.5 | 10.2 ± 16.8 | 10.6 ± 15.7 | |
| Thermo Scientific 49i | O$_3$ (ppbV) | 0.07 ppbV | 38.9 ± 8.8 | 39.2 ± 12.1 | 29.9 ± 11.9 | 31.2 ± 11.1 | Read et al. (2008) |
| Ocean Optics QE650000 | $j$NO$_2$ ($10^{-3}$ s$^{-1}$) | 15 % | 7.8 ± 2.7 | 9.3 ± 2.2 | 8.9 ± 2.5 | 8.7 ± 2.4 | See supplementary |
| | $j$O($^1$D) ($10^{-5}$ s$^{-1}$) | 15 % | 1.7 ± 1.2 | 3.0 ± 1.3 | 2.6 ± 1.2 | 2.6 ± 1.2 | |
| Picarro | CO (ppbV) | 1.0 ppbV | 99.0 ± 20.2 | 103 ± 17 | 80.0 ± 19.3 | 84.5 ± 16.6 | Zellweger et al. (2012, 2016) |
| | CH$_4$ (ppbV) | 0.3 ppbV | 1916 ± 26 | 1914 ± 29 | 1886 ± 34 | 1896 ± 30 | |
| GC-FID | Ethane (pptV) | 5.2 % | 1438 ± 600 | 1204 ± 608 | 518 ± 267 | 660 ± 449 | R. Steinbrecher (2019) |
| | Ethene (pptV) | 5.0 % | 31.2 ± 18.6 | 23.2 ± 9.8 | 27.5 ± 15.1 | 28.9 ± 19.6 | |
| | Acetylene (pptV) | 10.7 % | 134 ± 86 | 86.9 ± 82.4 | 22.6 ± 22.2 | 38.1 ± 38.5 | |
| | Propane (pptV) | 5.6 % | 336 ± 259 | 148 ± 195 | 20.6 ± 18.7 | 71.0 ± 133 | |
| | Propene (pptV) | 6.9 % | 8.6 ± 8.6 | 8.8 ± 11.5 | 8.0 ± 6.2 | 7.2 ± 6.1 | |
| | Iso-butane (pptV) | 6.4 % | 40.4 ± 39.5 | 11.0 ± 20.0 | 3.2 ± 4.3 | 8.4 ± 15.5 | |
| | n-butane (pptV) | 5.0 % | 82.8 ± 80.7 | 19.4 ± 36.0 | 6.0 ± 7.3 | 22.1 ± 40.5 | |
| | Iso-pentane (pptV) | 4.6 % | 11.1 ± 14.9 | 3.6 ± 6.2 | 5.2 ± 9.5 | 4.0 ± 6.7 | |
| | n-pentane (pptV) | 6.4 % | 8.7 ± 11.4 | 2.9 ± 4.7 | 1.7 ± 2.6 | 3.5 ± 5.2 | |
| | Benzene (pptV) | 4.8 % | 40.1 ± 30.5 | 22.9 ± 23.3 | 11.1 ± 10.5 | 17.3 ± 11.5 | |
| | Toluene (pptV) | 6.3 % | 4.6 ± 5.4 | 3.0 ± 4.2 | 2.9 ± 2.8 | 3.4 ± 3.1 | |
| | Methanol (pptV) | 20.7 % | 486 ± 563 | 698 ± 734 | 677 ± 603 | 857 ± 655 | |
| | Acetone (pptV) | 12.2 % | 506 ± 263 | 614 ± 274 | 767 ± 332 | 681 ± 213 | |
| Campbell Scientific weather station | Temperature (°C) | 0.4 °C at 5-40 °C | 22.0 ± 2.3 | 21.7 ± 1.4 | 24.5 ± 2.5 | 25.8 ± 2.1 | Carpenter et al. (2010) |
| | Pressure (hPa) | 1.0 hPa at 0-40°C | 1016 ± 4 | 1016 ± 3 | 1015 ± 4 | 1014 ± 3 | |
| | Relative Humidity (%) | 2 % at 10-90 % | 74.9 ± 12.8 | 77.2 ± 10.4 | 82.8 ± 8.8 | 81.1 ± 11.9 | |
| | Solar Radiation (W m$^{-2}$) | 5% | 615 ± 312 | 785 ± 251 | 737 ± 283 | 716 ± 273 | |

[a]Midday (12.00-15.00 UTC, local +1) mean ± 2σ for July 2017 – June 2020. [b]For further information on the instrument and the data processing.







**Table 3: Potential sources of NO₂ artefacts at the CVAO.**

| | ACS at 380 nm ($10^{-20}$ cm²)[a] | ACS at 385 nm ($10^{-20}$ cm²)[a] | ACS at 390 nm ($10^{-20}$ cm²)[a] | Conversion efficiency (%)[b] | Measured at the CVAO at midday (pptV)[c] | Modelled by GEOS Chem at midday (pptV)[c] | Potential artefact (pptV) |
|---|---|---|---|---|---|---|---|
| $NO_2 \xrightarrow{h\nu} NO$ | 59.24 | 59.42 | 62.0 | 50 | - | - | - |
| $BrONO_2 \xrightarrow{h\nu} NO_2$ | 3.85 | 3.37 | 2.97 | 2.8 | - | 0.5-1.5 | 0.014-0.042 |
| $ClONO_2 \xrightarrow{h\nu} NO_2$ | 0.121 | 0.137 | 0.091 | 0.1 | - | 0.5-1 | 0.0005-0.001 |
| $ClNO \xrightarrow{h\nu} NO$ | 8.86 | 7.82 | 6.86 | 6.6 | - | - | - |
| $ClNO_2 \xrightarrow{h\nu} NO_2$ | 0.3593 | 0.2687 | 0.2008 | 0.2 | - | ~0 | - |
| $BrNO_2 \xrightarrow{h\nu} NO_2$ | 17 | 17 | 16 | 14.3 | - | ~0 | - |
| $HONO \xrightarrow{h\nu} NO$ | 9.2 | 14.5 | 2.4 | 6.3 | 3-5 | 0.2-0.4 | 0.38-0.63 |
| $PAN \xrightarrow{\Delta} NO_2$ | - | - | - | ~5 | < 6 | ~20 | < 0.3 |
| **Total** | - | - | - | - | - | - | **0.69-0.97** |

[a]All absorption cross sections have been reported by IUPAC (Atkinson et al., 2004). [b]The reported conversion efficiencies have been calculated based on a NO₂ CE of 50%. [c]Midday is defined as 12.00-15.00 UTC (local+1).






**Table 4: Summary over the required additional artefact, RO₂, and XO to give [NO₂]obs./[NO₂]PSS ext. = 1 given as 50th (25th-75th) percentile when subtracting a NO₂ artefact of 0.7 pptV.**

| $\frac{[NO_2]_{obs.}}{[NO_2]_{PSS\ ext.}}$ | [CO] < 90 ppbV | 90 ppbV < [CO] < 100 ppbV | [CO] > 100 ppbV |
|---|---|---|---|
| | 1.00 (0.76 - 1.29) | 1.14 (0.89 - 1.47) | 1.42 (1.12 - 1.68) |
| Required additional artefact (pptV) | 0.00 (-2.65 - 1.70) | 1.9 (0.92 - 5.27) | 4.4 (0.95 - 9.27) |
| **Case I: Using BrO = 2.5 pptV and IO = 1.4 pptV** | | | |
| Required RO$_x$ (pptV)[a] | 49.45 (16.18 - 87.63) | 65.0 (33.68 - 112.5) | 109.7 (63.14 - 149.5) |
| Modelled RO$_x$ (pptV) | 48.89 (46.01 - 53.35) | 45.60 (35.69 - 54.71) | 44.99 (37.31 - 54.70) |
| Required RO$_2$ (pptV)[b] | 31.77 (-1.79 - 69.99) | 47.53 (16.81 - 93.93) | 90.49 (45.04 - 128.5) |
| Modelled RO$_2$ (pptV) | 33.66 (30.07 - 34.43) | 29.89 (21.50 - 36.32) | 27.62 (20.93 - 35.42) |
| Missing RO$_2$ (pptV)[c] | -0.25 (-31.85 - 39.69) | 20.19 (-14.23 - 66.44) | 61.33 (18.53 - 104.3) |
| **Case II: Using modelled RO$_2$ and HO$_2$** | | | |
| Required XO (pptV)[d] | 3.72 (-7.94 - 18.55) | 11.31 (-1.46 - 28.46) | 26.58 (10.70 - 42.52) |
| Missing XO (pptV)[e] | -0.18 (-11.84 - 14.65) | 7.41 (-5.36 - 24.56) | 22.68 (6.80 - 38.62) |

[a]Calculated using equation (IV). [b]Calculated using equation (VIII). [c]Calculated using equation (IX). [d]Calculated using equation (V).
[e]Subtracted 3.9 pptV of XO from the required XO (2.5 pptV BrO + 1.4 pptV IO).



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
