# Peer review of "Fundamental Oxidation Processes in the Remote Marine Atmosphere Investigated Using the NO NO2-O3 Photostationary State"

_Atmospheric Chemistry and Physics, 2022_

## Author Comment (AC1)

We thank all the Reviewers for their careful and considered comments, and have responded point by point below.

**Reviewer 1:**

The authors present analysis using three years of seemingly superb measurements from an excellent measurement site. The conclusions drawn from measurements of NO, $NO_2$, jNO2, O3, CO, and several VOCs is that there are "missing oxidants" that convert NO to $NO_2$ in the air, and that are not accounted for by past peroxy radical measurements. I recommend it be published after the following major and minor concerns are addressed:

1. The detailed model used is only as good and accurate as the inputs (i.e., compound concentrations constrained by measurements), and as impressive as the long-term dataset is, it does not include oxygenated VOCs. As such, it does not seem fair to expect that the models could accurately simulate the actual photochemistry given that it is likely not adequately constrained. Please include a discussion of the impact of unmeasured VOCs, especially oxygenated VOCs. Also please be clear what is meant by the term "missing" – are there reactions missing in the chemical mechanisms?

The long-term dataset does include the OVOCs methanol and acetone as shown in table 2 (and the model has been constrained to these). These are expected to be the dominant OVOCs at Cape Verde together with acetaldehyde. Unfortunately, we do not have reliable station measurements of acetaldehyde or formaldehyde currently, but measurements in the vicinity of the CVAO are available from previous short campaigns. Acetaldehyde from the ATom aircraft campaigns in October 2017, May 2018, and August 2018 show levels of between ~150 and ~250 pptV. Formaldehyde during the RHaMBLe campaign varied from 350 to 550 pptV (Mahajan et al. 2011). We have added a sensitivity study to the manuscript demonstrating the impact of constraining acetaldehyde to 150 pptV (lines 321-332) and formaldehyde to 450 pptV, compared to using the levels generated by the box model of ~8 pptV and 270 pptV, respectively. Total $RO_x$ levels increased by 3% from 52.7 pptV to 54.4 pptV, thus, the conclusions made in this study are not changed. Overall therefore, we believe that our model is adequately constrained. Of course it is possible that other more complex OVOCs or VOCs that are not typically measured, or below current detection limits, were present and contributed to $RO_2$ production or loss: these will contribute to the unaccounted-for/ additional $RO_2$ that are the focus of the manuscript.

The authors agree with the reviewer that "missing $RO_2$" was not the best term to describe $RO_2$ not accounted for in the box model. This has been changed throughout the manuscript to "unaccounted $RO_2$".

2. The analysis needs a more quantitative handing of the uncertainties. In particular, what is the uncertainty of the calculated quantity $[NO_2]$PSS-ext? (based on its constituent parts in equation III). For example, in line 343 of the manuscript. See also another comment below regarding the stated measurement uncertainties in Table 2 which require improvement. In numerous places it refers to older peroxy radical measurements and explains that those measurements are highly uncertain, especially at high RH. What are those uncertainties – both as stated in the original papers, and as concluded by the authors today?

The uncertainty of [NO2]PSS ext was used in the error bars in figure 4, however, it was not explicitly described in the manuscript. We have now added a description to the text (lines 363-365 and 373-376). The uncertainties in Table 2 are addressed below. The stated uncertainties for the measured peroxy radicals by the respective publications were already mentioned and discussed in the results section, but text has now been added to the introduction as well.

**Detailed comments**

Abstract

Line 29 "…implying 18.5-104 pptV (25th-75th percentile) of missing RO2 radicals" - the term "missing $RO_2$ radicals" is unclear. Please clarify as "…of $RO_2$ radicals missing from photochemical models".

The sentence has been rephrased as suggested.

Line 32: "If the missing RO2 radicals have an ozone production efficiency equivalent to that of…" The term ozone production efficiency is traditionally defined as the number of ozone molecules produced per NOx molecule. Please use a more accurate and defined term for what you mean in the abstract.

We have now changed the text to "If the additional $RO_2$ radicals inferred from the PSS convert NO to $NO_2$ with a reaction rate equivalent to that of …"

Line 34 (same sentence): "then the calculated net ozone production including these additional oxidants is similar to that observed"

The term "net ozone production" is unclear. Do you mean net ozone production rate (ppb/hr)? or does it mean "net ozone produced", which would be in # of molecules, or possibly mixing ratio (ppb)? Furthermore, it is confusing to refer to the "observed" ozone production rate, since nowhere in the abstract is it explained how that was "observed". Does "observed" actually man "calculated based on measured quantities"? Please clarify.

We mean the calculated net ozone production rate, which has now been made clear in the text.

Line 37 "and that measured and modelled RO2 are both significantly underestimated under these conditions." This is the first reference in the abstract to measured $RO_2$ and as such is quite confusing. Later in the paper it becomes apparent that it is referring to past measurements of $RO_2$ at this site. Please clarify.

The comparison to measured $RO_2$ has been removed from the abstract.

Body of manuscript

54: "Under very polluted conditions, where O3 is the only oxidant converting NO to NO2" – I disagree with that statement. There are plenty of very polluted conditions in which there are plenty of peroxy radicals present that also convert NO to $NO_2$ (e.g., Mexico City, Los Angeles…). This would be better phrased as "Under conditions in which O3 is the only oxidant converting NO to $NO_2$, …" and can clarify that perhaps the are referring to time periods with low sunlight and very high NO (I assume)

The sentence has been rephrased as the reviewer suggested.

72: the equations would be much easier to read if more subscripts were added. i.e., rather than $jNO_2[NO_2]$, write as $j_{NO2}[NO_2]$

We prefer the former ($j$NO$_2$) because it allows the "2" to be subscripted. But we will take editorial advice on this.

86: "However, PSS-derived ROx concentrations are generally higher than both measured and modelled values in rural conditions" – the wording can be tricky and sometimes confusing. The term "modelled" is confusing, since use of the PSS to derive ROx concentrations is in itself a simple model.

It has now been clarified throughout the text where "modelled" refers to either box-models, global models or steady state calculations.

116-117: "However, more recent instruments use "cavity absorption phase shift (CAPS)" - that should be attenuated rather than absorption, and probably wise to add "spectroscopy" or "spectrometry" afterwards.

Absorption has been changed to attenuated and spectroscopy has been added afterwards as suggested.

124: "… the increase in HO2 wall loss on wet surfaces" – humid surfaces, not wet surfaces. "Wet" implies there is a fair amount of liquid water on the surface (rather than a possible thin layer of adsorbed water).

"Wet" has been exchanged for "humid".

Lines 123 onward describe in detail the sensitivity of chemical amplifiers to humidity and specifics of the $RO_2$ being sampled. It appears that the main point of this section is to point out that these measurements are not perfect and subject to uncertainties. This is true of course, just as it is for measurements of all compounds. The resulting concentrations and stated uncertainties produced by chemical amplifiers ideally reflect the issues discussed in the text (RH dependence, dependence on organic nitrate and nitrite formation…). I recommend that this section describing $RO_2$ measurements by chemical amplifier conclude with a summary of the uncertainties of those measurements as described in the referenced papers. If the authors feel that the measurements are even more uncertain, they should state so explicitly. This might be especially important given that the peroxy measurements were made over 20 years ago.

The last sentence of the paragraph could easily be left off, since similar statements apply to all analytical measurement techniques: "It is therefore important to determine the optimal concentrations of reagent gas for each individual instrument as it could vary with what material has been used in the reactor". Similarly, it is important for each chemiluminescence instrument to use the proper ozone concentrations and flow rates, and for HOx LIF instruments to operate with the correct laser settings, NO flow rates….etc.

Thank you for these suggestions. A sentence on the uncertainties estimated by the mentioned studies has been added to the paragraph and the last sentence of the original text has been removed.

141: "The production of O3 (P(O3)) can be calculated using equation (VI)" insert the word rate after production

Done.

157: "In regions where the net O3 production is negligible or negative" again this is ambiguous wording, especially in light of the above note regarding the same term "net O3 production" (line 34). Please define what is meant by "net O3 production" – the rate? The change in O3 concentration over time?

The wording has been changed to the net O3 production rate.

Line 159 and 177: O3 should be [O3], or written as "O3 concentration"

Changed as suggested.

Line 180: define what is meant by "photochemical regime".

It has been clarified that it is whether the photochemical regime produces or destroys O3.

181 onward, and Table 1: Although later in the text the authors do a good job evaluating the possible interferences in the Chemi-photolytic converter technique, it is noteworthy that all almost all of the $NO_2$ measurements from Table 1 were made with chemiluminescence and a photolytic converter. The only study that used cavity ring-down spectroscopy (Tadic et al. 2020) appeared to find agreement between ROx(PSS) and ROx(model).

It has been specified that Tadic et al. reported a median ratio of 1.05, however, their data had a lot of variability.

181 – 189: "The large uncertainties associated with ROx measurements, especially at high humidities…" again, the authors really need to include the stated uncertainties from the chemical amplifier measurement papers themselves, and if they believe that the true uncertainties are higher, then they should state so. By how much higher would the uncertainties need to be to have agreement with ROx(model) or ROx(PSS)? Furthermore, is "high humidity" defined as greater than 80%, say, or greater than 50%? What is the range of humidity values observed during daytime at this site?

The uncertainties have now been stated earlier in the introduction, where the uncertainties due to humidity changes have also been explained in detail.

Table 2: The "accuracy" column is very confusing. For NO, $NO_2$, O3, CO, and CH4 an absolute mixing ratio is listed (e.g., 4.4 ppt), but for all the VOCs, a percentage is listed. The NO and $NO_2$ values undoubtedly need an accuracy listed in percentage, presumably determined largely by the calibration methods. Perhaps the 1.4 ppt and 4.4 ppt for the NO and $NO_2$ are actually the 1 sigma precision values? For what time averaging interval? The value for O3 seems erroneously low – 0.07 ppb! Please fix. The uncertainty of these measurements is crucial given their use in equations II and III.

It has now been made clear in the table that the hourly accuracies are 2 sigma. All compound accuracies are now stated in percentages.

The precision of the $O_3$ measurements is stated to be 1 ppbV for 1 second data from the manufacturer, which corresponds to a lower limit of 0.02 ppbV for hourly averaged data. Our zero measurements performed on the instrument in Cape Verde show a precision of 0.07 ppbV for hourly averaged data and it is this measured value which we report.

Section 3.1.1: given the detailed treatment of the $NO_2$ measurement artefact, it would be useful to include either a spectrum of the blue LEDs or to simply state its spectral width (FWHM).

The spectral width of the LEDs has now been added.

Line 261-262: "If NO2 is the product then it will be photolysed to NO with the same efficiency as NO2 in the ambient air" This does not seem correct, as for an interfering compound it's a two-step process and thus the $NO_2$ formed will have less exposure time to the UV radiation (e.g., X --> $NO_2$ --> NO, rather than $NO_2$ --> NO). An interfering compound that is converted to $NO_2$ in the photolysis cell should have a lower efficiency at making NO than $NO_2$ does.

We agree with the reviewer that photolysis of NO2 produced from an interference will have a lower efficiency, however, that efficiency is not known. The conversion efficiency of NO2 is therefore used to estimate an upper limit for the artefact. The text has been amended to explain that more clearly.

264: "Organic nitrates, HNO3, and NO3 do not photolyse at 385 nm and have therefore not been included in the evaluation of photolytic artefacts" Is this true for all organic nitrates?! There are many kinds – alkyl nitrates, hydroxy-alkyl nitrates, peroxy acyl nitrates…

It is correct that other kinds of organic nitrates not investigated in this study could photolyse at 385 nm. We have added text to described which organic nitrates have been considered here.

Line 273: "making it highly likely that a significant fraction of HONO is lost on the manifold before the air is introduced to the NOx instrument due to the high surface reactivity of HONO (Pinto et al., 2014)" What is the manifold made of? Glass? Teflon? If it's Teflon, then the quoted section seems like an overstatement. Have loss rates of HONO on surfaces been presented in other studes? Pinto et al 2014 appears to have little to say about surface losses and does not conclude that surface losses played a big role in that comparison study.

The manifold is made out of glass, which is now stated in the text. The Pinto et al. paper states in the introduction that "HONO is highly reactive on surfaces". Syomin and Finlayson-Pitts (2003), which has now been added to the text, demonstrated that loss of HONO in a glass chamber occurs, and found a decrease in the decomposition rate with increasing relative humidity.

331: Both of the references which provided the RO2+HO2 measurements by chemical amplifiers (Hernández et al., 2001 and Burkert et al., 2001) were from 21 years ago. Do changes in background NOx and O3 affect the context of their inclusion in figure 2?

The references for ROx are indeed over 20 years old, however, they are only used for comparison to the box model. Both NOx and O3 have been measured at Cape Verde since 2006. No significant increases in either have been observed during the past 15 years.

343: "Daily midday values of [NO2]PSS ext were calculated using equation III" What is the combined uncertainty of [NO2]PSS ext? Note that this is an important area where the uncertainties of the past chemical amplifier measurements can be addressed quantitatively, as it is part of equation III. This is a crucial area of revision.

The combined uncertainties of [NO2]PSS and [NO2]PSS ext. have now been added to the text. The uncertainties of the past PERCA measurements are discussed later in the paper, where we believe it is more suitable.

Line 361: "the abundance of NO on …" although the term "abundance" is commonly used synonymously with "concentration", I advise against it in this case as NO molecules were anything but abundant!

Abundance has been exchanged for mixing ratio.

**Reviewer 2:**

Andersen et al. use long-term (years) measurements of $NO_x$, $O_3$, organic compounds and associated parameters from a remote marine sampling location to evaluate understanding of radical chemistry affecting $NO_2/NO$ ratios and ozone production. This topic is of wide interest as radical chemistry is central to understanding global oxidation processes, and many studies have failed to explain the observed $NO_x$ partitioning in a variety of chemical environments. Strengths of the work are uniqueness of the dataset, and analysis using GEOS-Chem and a detailed chemical box model to evaluate the chemistry.

Overall, I think the paper is well written, provides an excellent review of and links to the prior work on this topic, and has interesting analysis. I think that the paper will deserve publication but that the authors should first consider a few important points concerning the limitations of the measurements and modeling analysis and how that might affect the way that the conclusions are stated.

General comments:

1) The primary conclusion of the paper is that the $NO_2/NO$ ratio observations are consistent with the expected $NO->NO_2$ oxidants in the cleanest conditions, but more polluted air masses would require significantly more organic peroxy radicals or halogen oxides to explain the observed $NO_2/NO$ ratios. This is first stated in the paragraph beginning on line 354. I am not convinced, however, that there is a clear difference in the behavior between the more pristine and more polluted air masses. In other words, it is not clear to me that one can say the cleaner data definitely are completely explained by the known chemistry whereas the more polluted data have a different behavior. I think a more thorough discussion of the uncertainties of each data point due to precision or artifact uncertainties would help the interpretation of the figures.

For example in Fig 3B while the scatter of data at NO2 < 20 ppt are hard to distinguish from the 1-1 line, I would not say by eye that the overall trend there is different than at the higher NO2 mixing ratios. In Figure 5, while enhancements in acetylene and ethane are associated with higher than expected NO2, the data with low acetylene and ethane do not cluster around a value of 1 for NO2_obs / NO2_pss, but appear to have significantly lower than expected NO2. I did not see discussion of the lower than expected $NO_2$ observations. In Figure 6 while the CO < 90 ppb data are centered around a value of 1 for NO2_obs / NO2_pss, many of the points are not close to one. Is the width of the histogram explained by the precision of the measurements or is it possible that some of the width here is also evidence for incomplete understanding of the chemistry?

The statement about the trend in Figure 3B being different for NO2 > 20 ppt has been replaced by a discussion of the linear fits of the data in figure 3A and B and the increase in the slope.

While it looks like the data in Figure 5 at low mixing ratios of acetylene and ethane do not cluster around 1, they actually do, but the high mixing ratio data plotted on top obscured this. The figure has been changed to show this more clearly and the text has been rephrased to fit with the new figure.

A student's t-test has been performed on the data with CO < 90 ppb and CO > 100 ppb to verify that the two groups are significantly different. This has been described in the text. The data where 90 ppb < CO > 100 ppb has not been tested as it is a transition between the two other categories.

2) As I understand it from this paper and Anderson et al. 2021, a potential positive artifact on the $NO_2$ measurement from the photolytic converter is assumed to be negligible (Anderson et al., 2021 state that measurements of zero air show 0 – 10 ppt of $NO_2$, which is assumed to be real $NO_2$ in the zero air). While I understand the problems/challenges with experimentally determining if there is a real surface artifact, I find it concerning that the potential for a positive artifact in the $NO_2$ measurement due to illumination of species on the walls of the photolytic converter is assumed to be zero. It is well documented that typically a positive NO signal of at least a few ppt will be generated by illuminating such converters (even quartz ones) even in the presence of synthetic, NOy-free air (e.g. Gao et al., 1994, Pollack et al., 2010, others). Can the authors please comment in the artifact section in some way on this? What would the impact be if there were a few ppt of fake $NO_2$ from the converter? Perhaps the lowest measured $NO_2$ could be used at least as an upper limit of such an artifact. Are there other upper limits that can be stated for such an artifact?

In our Andersen et al. 2021 paper we show comparable results from two different photolytic converters when assuming the quartz converter has a zero artefact (within uncertainty). In this study, we show that for very clean air the photostationary state of $NO_x/O_3$ can be explained by 0.7 pptV of NO2 artefact. This artefact agrees well with our calculated average artefact of 0.67-0.95 pptV from known interferences, as shown in Table 3, and is also not distinguishable from zero within uncertainty. We use 0.7 pptV as the NO2 artefact in the remainder of the paper. If the artefact is higher than 0.7 pptV, then we would predict more NO2 than we measure for very clean air, i.e. the PSS ratio would be less than 1. If we would

use the lowest measured $NO_2$ (2.3 pptV) as the artefact, then we would have times with zero $NO_2$, which is not possible.

Text has been added to the artefact discussion to clarify these points.

3) I suggest that the authors put a bit more emphasis/discussion on the good agreement shown in Fig. 7 between measured and calculated ozone tendency. It could be argued that this is more important than being able to reproduce the $NO/NO_2$ ratio, and therefore remaining uncertainties or discrepancies in observed vs calculated $NO_2/NO$ are less important to resolve since the ozone tendency seems nicely explained.

We agree with the reviewer that the good agreement between measured and calculated ozone tendency is an important feature of this work. However, in this very low NOx environment, the ozone tendency is quite insensitive to changes in peroxy radical concentrations (as discussed in lines 556-559). If there is some fundamental missing understanding however of peroxy radicals, this could clearly have a large impact in more polluted regions. We have added such a statement on line 564-565.

Specific comments by line:

Line 60: Suggest defining $RO_2$ as 'organic peroxy radicals' rather than just 'peroxy radicals.'

"Organic" has been added to the text.

252: Recommend using the symbol s rather than defining the ACS acronym.

The ACS acronym has been used to avoid using sigma for both standard deviations and absorption cross section.

260: Can you state the width of the LED spectrum?

The width has been added.

277: While GEOS-Chem may not show a coherent seasonal pattern for NOy, clearly there is a lot of real variability that is likely related to airmass origin, and higher NOy is probably related to pollution sources. PAN for example could matter. Could you comment on the origin of the variability in GEOS-Chem? Perhaps adding a timeseries of CO to Fig. S7 would be helpful.

Both a modelled and measured timeseries of CO have been added to figure S7, which shows that the variability in NOy does not depend on the seasonality in CO.

292: The GEOS-Chem timeseries of PAN (S7) which seems to be routinely above 20 ppt would suggest that if GEOS-Chem has some skill here the PAN would be above this 6 ppt detection limit frequently, or always. Can you comment on this?

We agree that GEOS-Chem predicts much higher PAN than the measured values, however investigating the reasons for this are outside the scope of this manuscript. The measured values are based on an established technique and so we use them here in the calculation of the artefact.

322: Since the calculation of RO2 is critical to the argument of the paper, it would be helpful to see more information about the relative importance of these measured $RO_2$ precursors. Is there any correlation between the calculated $RO_2$ and the pollution indicators? Do the authors think that the missing $RO_2$ sources could be due to VOCs that are not measured by the GC system at CVAO? If the air is of African origin and possibly influenced by biomass burning, can the authors comment on how sufficient the measured suite of VOCs might be in comparison to recent those reported in more recent papers with comprehensive measurements of biomass burning VOC emissions? Overall, I'm a bit unsure if 'missing' is the right word to use to describe the unaccounted for $RO_2$, or rather that we should expect there are a number of important organic compounds that were not measured.

The box modelled $RO_2$ shows a strong correlation with the measured $jO^1D$, but no correlation to CO (pollution tracer) or $CH_4$ (expected dominant source of RO2 in the marine environment). A sentence has been added.

The authors agree that "missing RO2" can be misleading and it has been changed throughout the text (see also response to reviewer 1).

[Figure]

460: I would say that the required additional factor for XO is higher than that of RO2 not because of the difference in rate coefficients, but because the measured/calculated XO is $<<$ measured/calculated RO2.

The description has been deleted from the text.

**Figures**

Fig1: please provide a colorscale and explanation. Does each point represent the calculated location of an air parcel 10 days prior to arrival at CVAO?

The back-trajectory model releases 1000 particles from the point of interest and calculates their latitude/longitude/altitude every 10800 seconds (3 hours), backwards over a 10-day period. In this figure, the locations of the 1000 particles at each timestep are totalled in a 1 degree x 1 degree grid, to show the density of particle distribution over the 10-day period. Grid boxes containing less than 10 particles over the 10-days are masked. The figure is meant as a visualisation tool, to demonstrate the seasonality of back-trajectory footprints reaching Cape

Verde. We have not provided a colour scale for this figure in the paper as we don't believe it adds to the understanding of the figure. The unit would be "n particles per grid square".

Fig2: Would be nice to mention in the caption the seasons of those campaigns.

The months of the different campaigns have been added to the caption.